

# Simulation of Fragmental Rockfalls Detected Using Terrestrial Laser Scans from Rock Slopes in South-Central British Columbia, Canada

Zac Sala[1,2], D. Jean Hutchinson[1], Rob Harrap[1]

[1]Department of Geological Sciences and Geological Engineering, Queen's University, Kingston, K7L 3N6, Canada
5  [2]BGC Engineering Inc., Vancouver, V6Z 0C8

*Correspondence to*: Zac Sala (zac.sala@queensu.ca)

**Abstract.** Rockfall presents an ongoing challenge to the safe operation of transportation infrastructure, creating hazardous conditions which can result in damage to roads and railways, as well as loss of life. Rockfall risk assessment frameworks often involve the determination of rockfall runout in an attempt to understand the likelihood that rockfall debris will reach an element 10  at risk. Rockfall modelling programs which simulate the trajectory of rockfall material are one method commonly used to assess potential runout. This study aims to demonstrate the effectiveness of a rockfall simulation prototype which uses the Unity3D game engine. The technique is capable of simulating rockfall events comprised of many mobile fragments, a common limitation of available rockfall modelling programs. Five fragmental rockfalls were simulated using the technique, with slope and rockfall geometries constructed from high-resolution terrestrial laser scans. Simulated change detection was produced for 15  each of the events and compared to the actual change detection results for each rockfall as a basis for testing model performance. In each case the simulated change detection results aligned well with the actual observed change in terms of location and magnitude. An example of how the technique could be used to support the design of rockfall catchment ditches is shown. Suggestions are made for future development of the simulation technique with a focus on better informing simulated rockfall fragment size and the timing of fragmentation.

20  ## 1 Introduction

Rockfall is a mass-movement hazard often found in mountainous environments, posing risk to human lives and the safe operation of infrastructure. In Canada, rockfall hazard is particularly problematic for transportation infrastructure, where traffic corridors have been constructed in steep river valleys, adjacent to natural and cut slopes. In many of these valleys, right-of-way space is limited, forcing operators to construct highways and railways with minimal clearance to potentially unstable rock 25  masses. The proximity and extensive presence of fractured and weathered rock slopes, combined with the seemingly unpredictable nature of rockfall events, makes it difficult for operators to manage this hazard.

Like any geohazard, we are interested in knowing how frequently we can expect events to occur, where they are most likely to come from, and whether or not they will reach and cause harm to our elements at risk. For rockfall this could involve identifying lithologies which are more susceptible to rockfall (e.g. Rosser et al., 2005; van Veen, 2016), or correlating rockfall



frequency to triggering events like severe rain storms (e.g. D'Amato et al., 2016; Pratt et al., 2018). In order to build these relationships, we first need knowledge of previous rockfall in the area. This is one of the main challenges in the study of rockfall. Due to challenging terrain in mountainous areas, and safety concerns where rockfall activity is high, site access to the slopes of interest is often limited, prohibiting direct measurement.

In order to circumvent these obstacles, modern remote sensing techniques, such as LiDAR, have seen widespread use in the study of landslide hazards like rockfall (Jaboyedoff et al., 2010). Terrestrial laser scanning (TLS) in particular has been increasingly applied in recent years to the study of rock slope instabilities (Abellán et al., 2014; e.g. Lato et al., 2009; Lato et al., 2015; Kromer et al., 2017). TLS provides a detailed 3D model of the slope geometry, supporting a range of geotechnical and geomechanical analyses. From a single scan we can extract slope angle measurements, map rockmass structure, and look

for evidence of past rockfall (Telling et al., 2017). Using multiple successive scans of the same slope, progressive changes to the slope surface can be measured. Multi-temporal scanning enables researchers to detect rockfall events over time and build detailed magnitude-frequency relationships for slopes, supporting the study of processes such as coastal cliff erosion (e.g. Rosser et al., 2007; Williams et al., 2018), as well as hazard and risk assessments for linear infrastructure such as railways (e.g. van Veen et al., 2017). The identification of rockfall events using sequential scans, also permits the back-analysis of

rockfall triggering factors and failure mechanisms. A study by Kromer et al (2015) used TLS change detection to investigate the occurrence of a 2600 m$^3$ failure above a section of the CN Railway in western Canada, including an analysis of structural constraints, pre-failure deformation, and pre-cursor rockfall leading up to failure.

    While the application of TLS to rock slope monitoring is often focused on determining how likely future rockfall is or where the rockfall may come from, it is also important to determine how likely it is that the rockfall material will reach an

element at risk should a fall occur. In order to answer that question, rockfall modelling programs can be used to simulate the runout of falling rock material using numerical models (Turner and Duffy, 2012). A rockfall simulation requires a representation of the slope surface and rockfall mass being modelled in 2D, 2.5D, or 3D. TLS can support rockfall simulation by providing a high-resolution 3D model of the slope surface, and in cases where a rockfall event has been identified using change detection between multiple scans, the volume and location of the rockfall mass. In order to make use of the quality and

quantity of 3D point cloud data being collected as part of a rock-slope monitoring program in western Canada, a 3D rockfall simulation technique using game-engine technology has been developed (Ondercin, 2016; Sala, 2018). The technique uses the video game development platform Unity3D (Unity Technologies, 2018) and is capable of simulating rockfall runout using fully 3D meshes built from TLS point cloud data.

    One of the strengths of this simulation technique is its ability to simulate rockfall runout using numerous interacting bodies.

This capability allows us to produce simulations of fragmental rockfall events, which are defined by the presence of multiple mobile fragments of rock during runout (Hungr and Evans, 1988). Conventional rockfall modelling programs (e.g. Rocfall (Rocscience Technologies, 2016), Rockyfor3D (Dorren, 2015), RAMMS:ROCKFALL (Bartelt et al., 2016)) simulate single boulder trajectories at a time and therefore are unable to model fragmental rockfall runout. The goal of the work presented in



this paper is to demonstrate the capability of our novel simulation technique to model a series of rockfall events detected using TLS change detection at two rock slopes in south-central British Columbia. Emphasis is placed on the ability of the technique to be used for fragmental rockfall runout simulation, including a discussion of how these types of simulation could support mitigation design.

**2 Study Sites**

The rock slopes of focus for this paper are part of the Ashcroft subdivision of the CN Railway. The subdivision follows sections of the Thompson and Fraser river valleys, located between the towns of Ashcroft and Lytton, British Columbia. This region serves as an important transportation corridor for Canada, with the presence of the CN Rail mainline, as well as sections of the Canadian Pacific Railway, and the Trans-Canada Highway. A previous study by Piteau (1977) identified that rock cuts in this

region are subject to slope instability issues due to a combination of lateral erosion from river activity, over-steepening from blasting during the construction of the railways, and a lack of adequate rockfall catchment areas.

Two sites in the Ashcroft subdivision will form the basis for this research, the White Canyon, and Goldpan. The location of these sites along the Thompson River, and proximity to the town of Lytton, BC, can be seen in Figure 1. The monitoring of these slopes is part of the Railway Ground Hazard Research Program, a collaborative research initiative focused on the

characterization and assessment of mass movement hazards affecting Canadian railways.



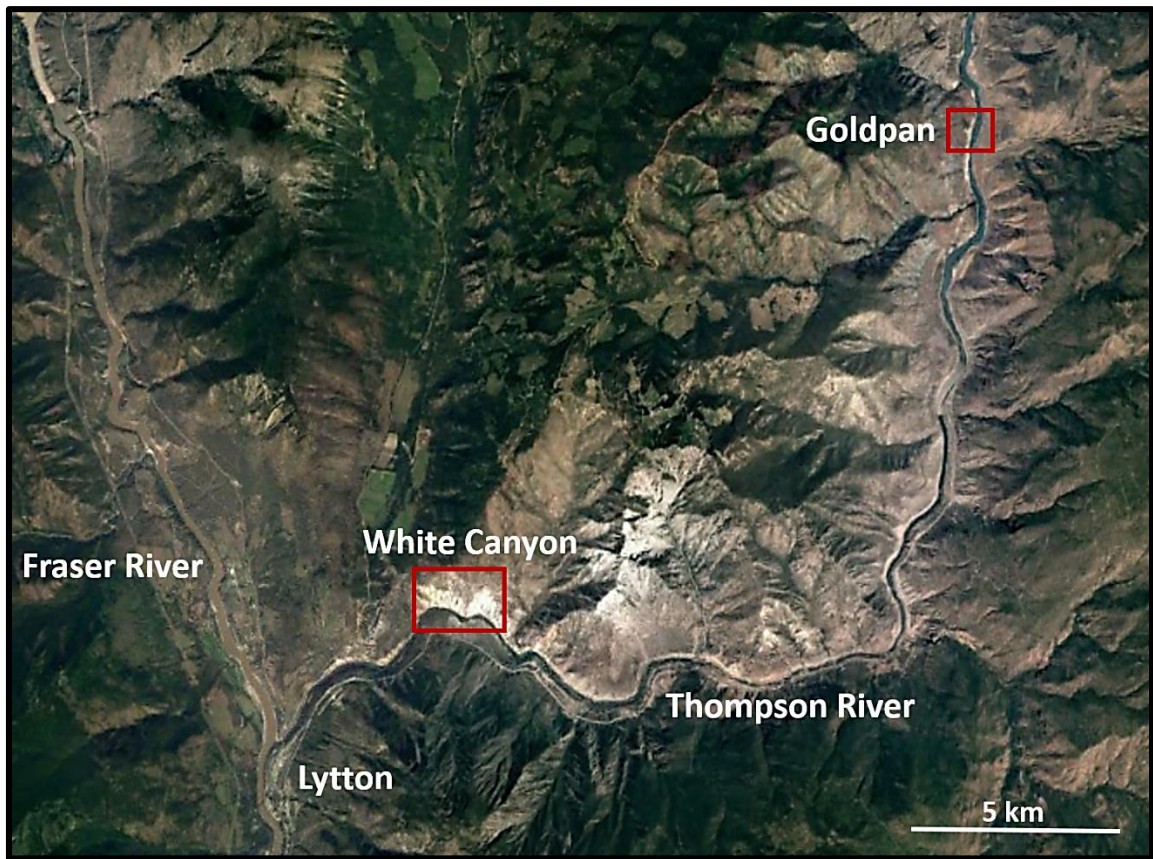

**Figure 1: Aerial imagery showing the location of the White Canyon and Goldpan field sites situated along the Thompson River, near the town of Lytton in south-central British Columbia. This region is approximately 160 km northeast of the city of Vancouver. (Image Source: Google Earth. Digital Globe 2018)**

Monitoring at both sites has been ongoing since 2012, with data collection taking place on a seasonal basis, approximately every 3 months. Point cloud data collection consists of TLS scans acquired using an Optech ILRIS 3D-ER (2012-2017) or Riegl VZ-400i (2018) LiDAR system. Additionally, ALS data coverage for both sites, with an average point spacing of 0.3 m, was acquired in 2014 and 2015. Site photos are collected using a Nikon D700 or similar camera with a 135 mm prime lens.

5    A series of photos of the slope are taken using a GigaPan EPIC Pro robotic camera mount, and stitched together into a single high-resolution site panorama using Gigapan Stitch (GigaPan Systems, 2013).

Goldpan is located on the north side of the Thompson River, approximately 26 km upstream from the town of Lytton. Present at the base of the slope is a section of the CN Rail main line. The rock slope spans 800 m, rising up to 65 m vertically above track level. The average slope angle at the site is 55-60 degrees. TLS data is collected from scanning vantage points

10   across the river from the slope, accessed via Goldpan Provincial Park. Scan distances range between 170 – 230 m, producing an average point spacing of approximately 6 cm.





The White Canyon is located on the north side of the Thompson River, approximately 5 km upstream from the town of Lytton. Present at the base of the slope is a section of the CN Rail main line. The rock slope spans 2.4 km, rising up to 375 m vertically above track level. The average slope angle at the site is 40-45 degrees. TLS data is collected from scanning vantage points across the river from the slope. Scan distances range between 400-600 m, producing an average point spacing of approximately 10 cm.

Goldpan and the White Canyon are located in the Intermontane belt of the Canadian Cordillera. Both sites belong to the Quesnellia volcanic arc terrane, which is composed of Carboniferous to Mid-Jurassic volcanic, sedimentary and plutonic rocks (Struik, 1987; Monger and Nokelberg, 1996). The rockmass at Goldpan is largely massive and is predominantly composed of volcanics of the mid-Cretaceous Kingsvale-Spences Bridge Group. The main rock types of this Group are basaltic to andesitic flows interspersed with volcaniclastic sandstones, shales and conglomerates (Brown, 1981). The comparatively large rock slope of the White Canyon belongs to the Mt Lytton plutonic complex (Greig, 1989). Locally, the dominant rock type is an amphibolite grade quartzofeldspathic gneiss. Amphibolite banding is also present and gneissic layering is cross cut throughout the canyon by intrusive phases of gabbro, tonalite, and granodiorite (Brown, 1981). Preferential weathering around these intrusions often results in the formation of vertical rock spires which act as source zones for rockfall.

Mitigative measures at both sites have been installed in response to frequent rockfall activity impacting railway operations. At Goldpan this includes rockfall wire mesh draped over parts of the slope, extensive sections of shotcrete, and four concrete rock sheds. In the eastern half of the White Canyon there are two timber rock sheds and one concrete shed, as well as wire mesh rockfall nets and concrete lock blocks adjacent to the track. The western half of the White Canyon also has wire mesh rockfall nets and lock block retaining walls, as well as an additional three concrete rock sheds, and one timber rock shed. Slide detector fences are present at both sites, comprised of horizontal wires strung between upright telephone poles, and provide warning by switching the track signal to stop when broken.

## 3 Rockfall Events

Five rockfall events were selected from a database of rockfalls which have been identified using change detection at White Canyon West, White Canyon East and Goldpan since 2012 (Kromer et al., 2015). These events have masses ranging from 2 m$^3$ to 170 m$^3$, exhibiting a combination of structurally controlled failure modes including wedge sliding, planar sliding, toppling, and overhanging blocks. Images of the five slope sections where the rockfall events of interest took place can be seen in Figure 2.





**Figure 2: Photos of five rock slope sections, prior to failure, from Goldpan (A) and the White Canyon (B-E) sites adjacent to the CN Railway. The red regions indicate the source zone for the rockfall events discussed in this chapter.**



Each event showed material accumulation below the source zone in the change detection results. This is essential information for comparison with our simulation results. Each event is also close to track level, with the highest fall occurring 46 m vertically above the track. These events were selected because the shorter distance from source to a notable accumulation of material presents a simpler trajectory for back-analysis and reduces potential confusion interpreting whether the material

5  gain is due to the selected rockfall or another mass movement nearby. In each case, change was detected using the Multiscale Model to Model Cloud Comparison (M3C2) point-point distance calculation (Lague et al., 2013) in CloudCompare (2018), between the pre- and post-fall TLS scans. A summary of each of the rockfall events including change detection results, and site photos before and after the event are provided in Figures 3 – 7. Discontinuity and slope angle measurements used for the stereonet failure mode analysis shown in each figure was completed using the Compass tool in CloudCompare.





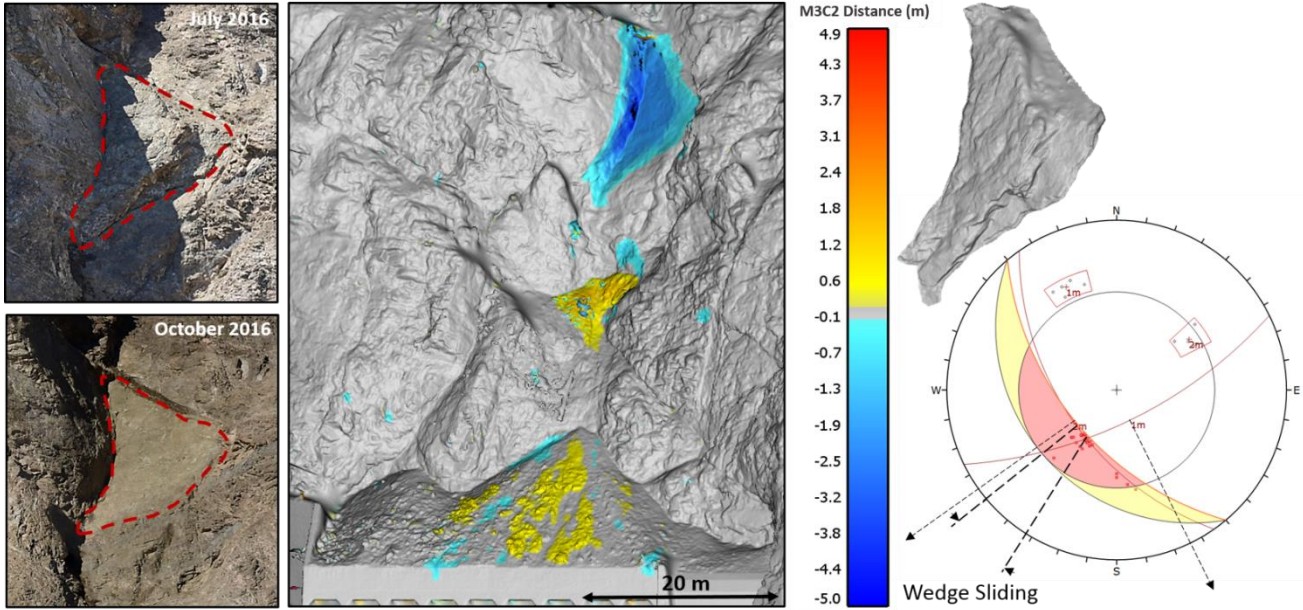

**Figure 3: Site A.** Visual overview of the 170 m³ wedge sliding rockfall event detected at the Goldpan site between July and October 2016. The event involved a large triangular slab of weathered and jointed rockmass which failed approximately 35 m above a rock shed which is protecting the track at Goldpan. Site photos (left) of the source zone pre- and post-failure are shown. Change detection results of the event are shown (center) with cool colours indicating material loss and warm colours indicating material accumulation. Material from the rockfall accumulated on the bench of the mid-slope gully, as well as on top of the rock shed, with the majority of the material running out over the shed and leaving the slope. The rockfall hull of the event, extracted from the pre- and post-failure meshes is shown (upper right), as well as a stereo-net representation of the wedge sliding failure mode.





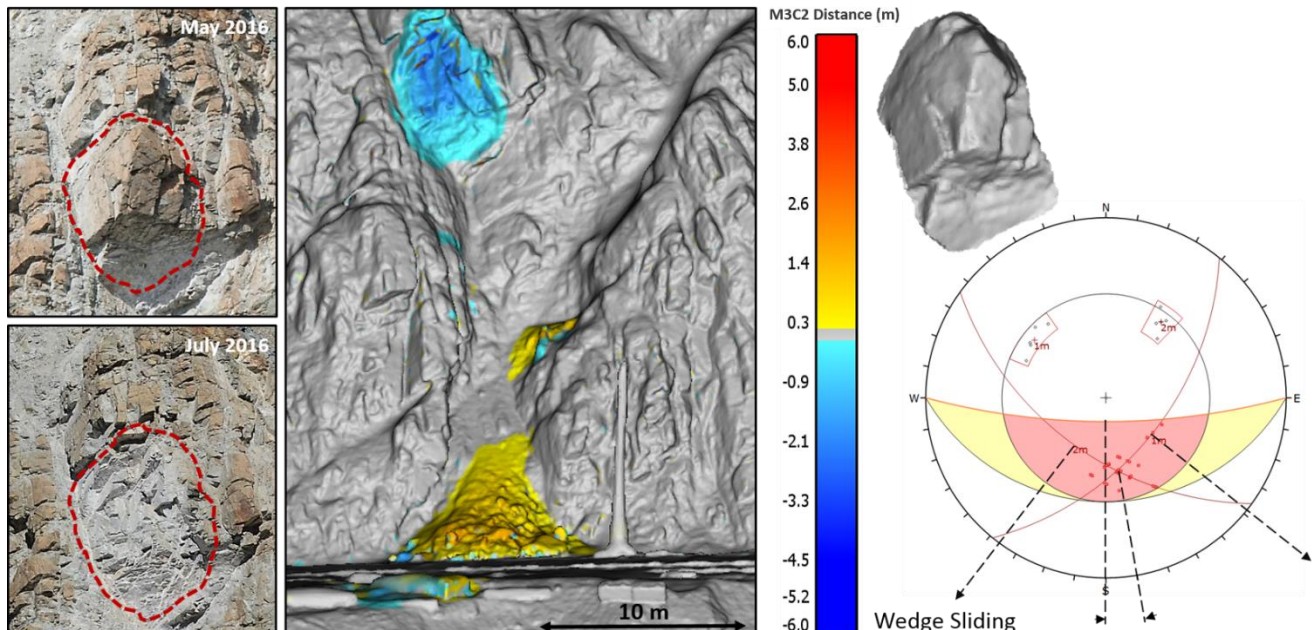

**Figure 4: Site B. Visual overview of the 24 m³ wedge sliding rockfall event detected at the White Canyon East site between May and July 2016. The event involved a large pseudo-cubic block of weathered and jointed rockmass which failed approximately 46 m above track level. Site photos (left) of the source zone pre- and post-failure are shown. Change detection results of the event are shown (center) with cool colours indicating material loss and warm colours indicating material accumulation. A small portion of the rockfall volume was retained in the gully leading down to track level, with the majority of accumulation taking place in the track-side ditch. The rockfall hull of the event, extracted from the pre- and post-failure meshes is shown (upper right), as well as a stereo-net representation of the wedge sliding failure mode. It should be noted that parts of the source rockmass also exhibited overhanging sections.**



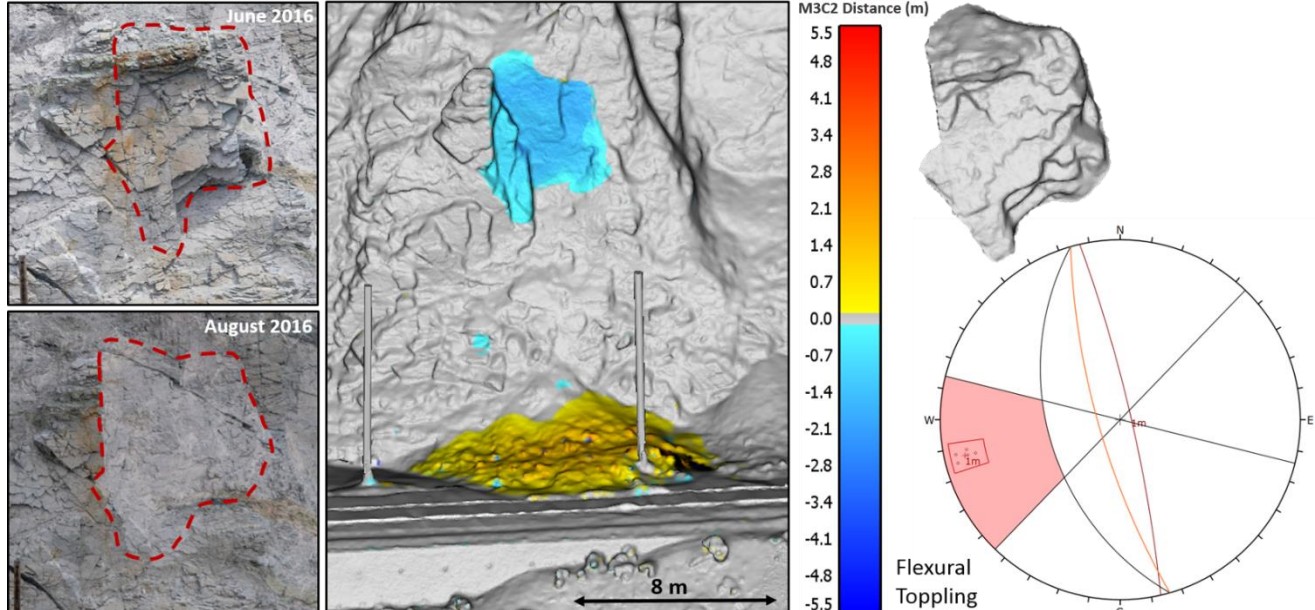

**Figure 5: Site C. Visual overview of the 32 m³ flexural toppling event detected at the White Canyon East site between June and August 2016. The event involved a large slab of weathered and heavily jointed rockmass which failed approximately 14 m above track level. Site photos (left) of the source zone pre- and post-failure are shown. Change detection results of the event are shown (center) with cool colours indicating material loss and warm colours indicating material accumulation. Change detection results indicate the bulk of material from the rockfall event was retained in the track-side ditch. The rockfall hull of the event, extracted from the pre- and post-failure meshes is shown (upper right), as well as a stereo-net representation of the flexural toppling failure mode. It should be noted that parts of the source rockmass also exhibited overhanging sections.**

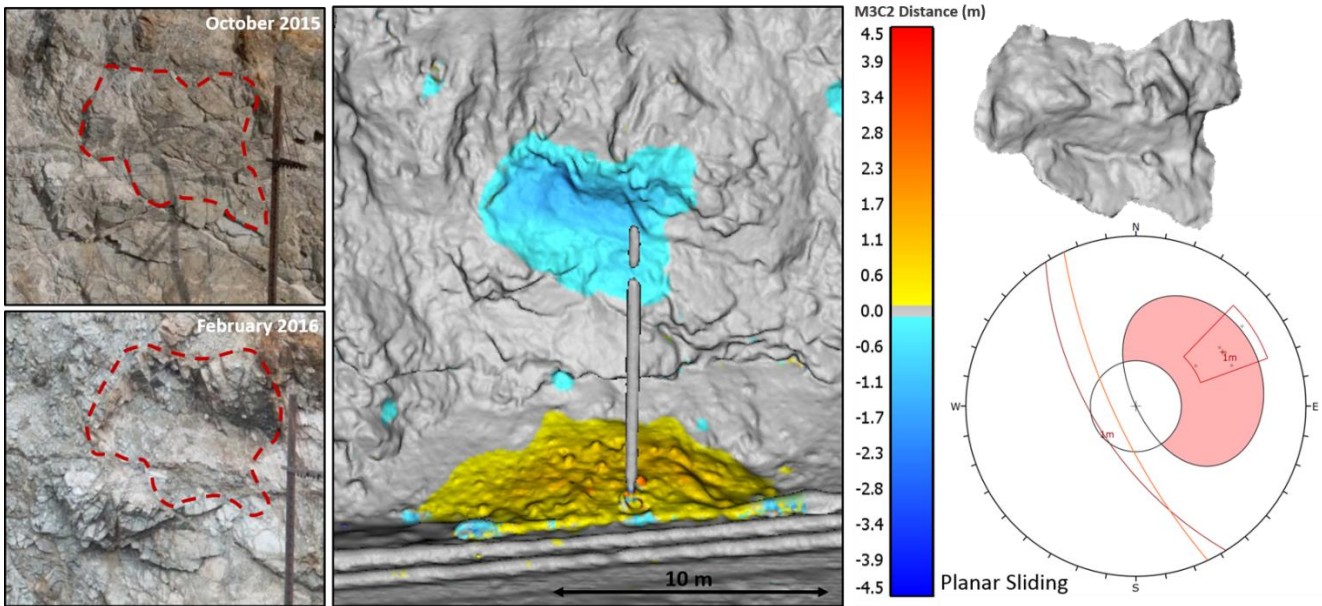

**Figure 6: Site D. Schematic overview of the 15 m³ planar sliding rockfall event detected at the White Canyon West site between October 2015 and February 2016. The event involved an irregularly shaped slab of weathered and heavily jointed rockmass which failed approximately 9.5 m above track level. Site photos (left) of the source zone pre- and post-failure are shown. Change detection results of the event are shown (center) with cool colours indicating material loss and warm colours indicating material accumulation. Change detection results indicate that the bulk of the material from the event was retained by the track-side ditch. The rockfall hull of the event, extracted from the pre- and post-failure meshes is shown (upper right), as well as a stereo-net representation of the planar sliding failure mode.**


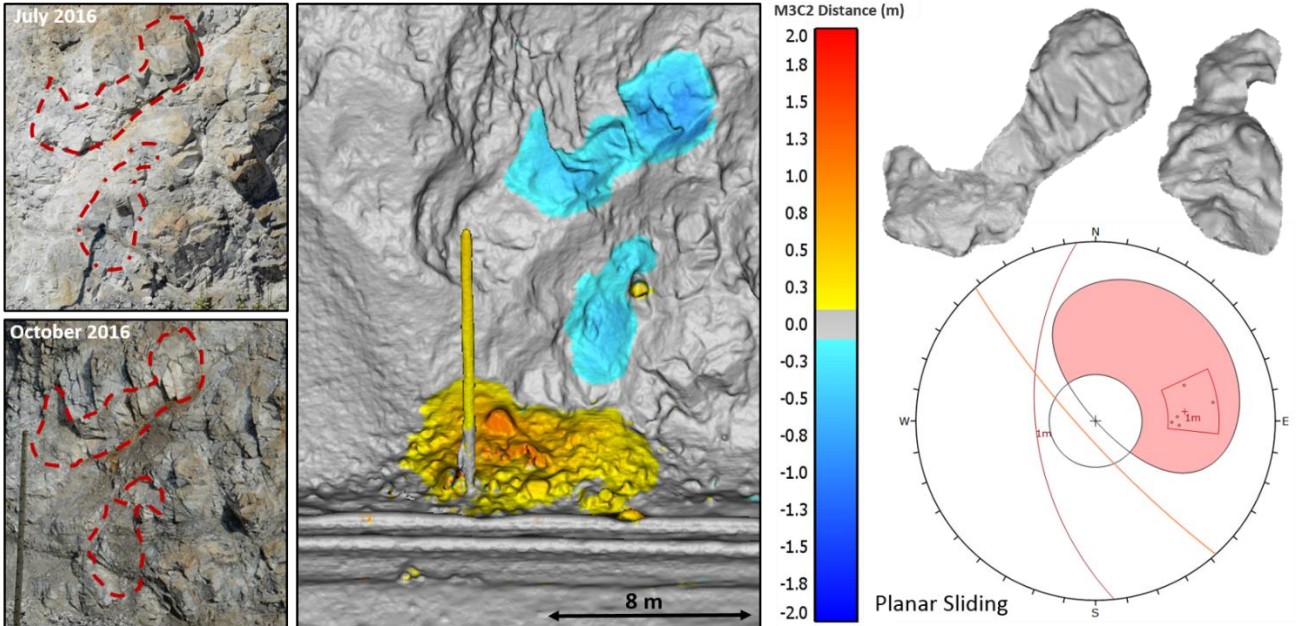

**Figure 7: Site E. Schematic overview of the 8 m³ planar sliding event detected at the White Canyon West site between July and October 2016. The event involved the failure of two distinct sections of the jointed source rockmass, with the upper (6 m³) and lower (2 m³) failures releasing from 10 m and 6 m above track level respectively. Site photos (left) of the source zone pre- and post-failure are shown. Change detection results of the event are shown (center) with cool colours indicating material loss and warm colours indicating material accumulation. Interpretation of the post-fall imagery indicates that the lower failure was inside the slide path of the upper rockfall event. It is our interpretation that these events likely occurred simultaneously, with the lower failure occurring as a result of impact from the upper failure. Change detection results indicate that the bulk of the material from the two events was retained by the track-side ditch. The rockfall hull of the events, extracted from the pre- and post-failure meshes is shown (upper right), as well as a stereo-net representation of the planar sliding failure mode for the upper, larger event.**

### 3.1 Event Fragmentation

While the selected rockfall events were not observed directly, each rockfall is believed to have been comprised of multiple mobile fragments. This interpretation is based on observations of the fractured state of the source zones pre- and post-fall, as well as the size and distribution of visible rock fragments in the post-fall areas of accumulation, below the source zone and adjacent to the track. An example of this can be seen in Figure 8 for the White Canyon overhanging wedge event. In these photos it is clear that the source rock mass is heavily jointed, and that the accumulation of material generated from the event is not a few large blocks but rather a deposit of coarse granular material.




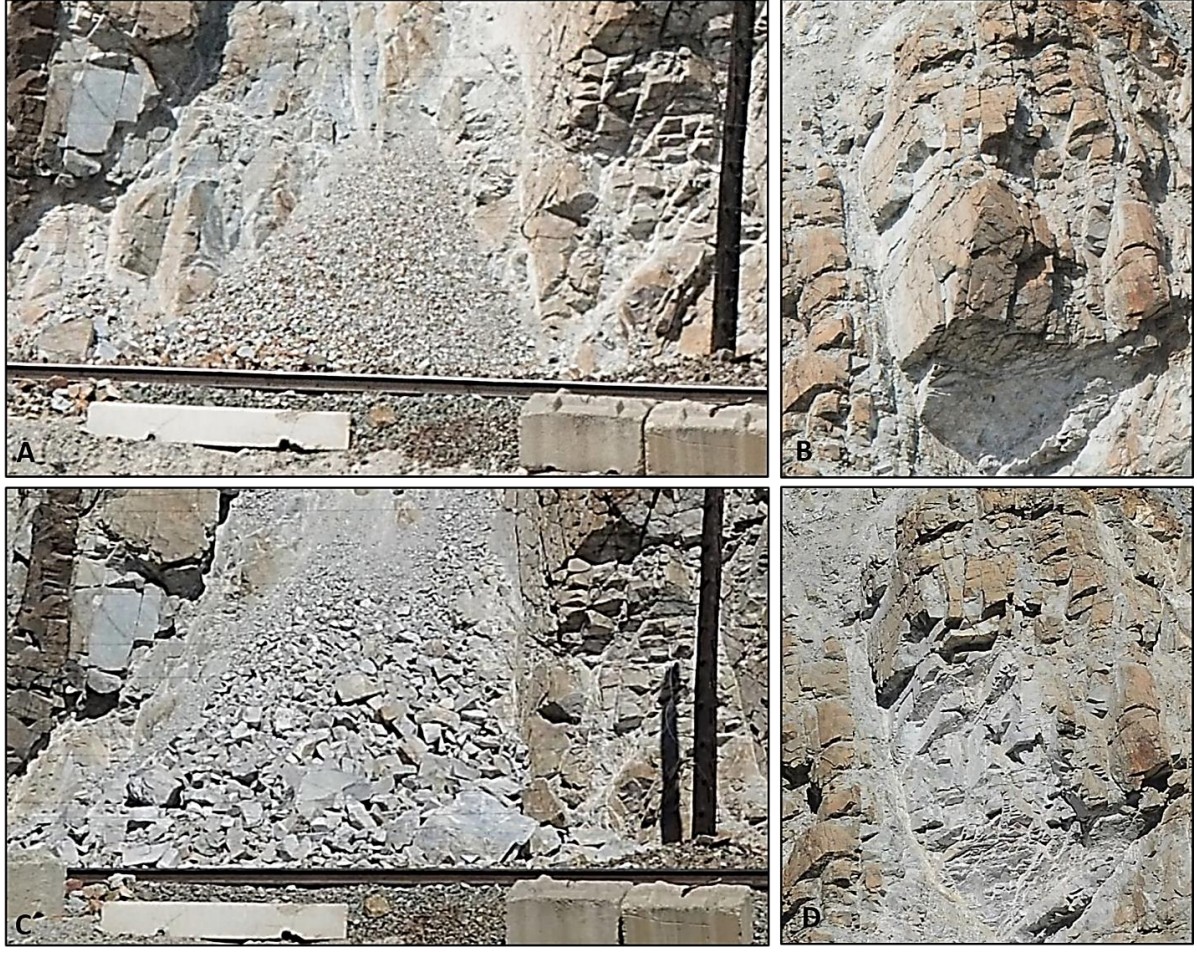

**Figure 8: Before and after photos of the 24 m³ wedge sliding event from White Canyon East, Site B. The top row of photos shows the pre-fall debris present in the track-side ditch (A), as well as the pre-fall source zone (B). The bottom row shows the post-fall accumulation in the track-side ditch (C), as well as the post-fall source zone (D). From these images we can see the heavily jointed state of the rockfall mass pre-fall and the rockfall back-scarp, post-fall. A notable increase in the quantity and size of debris fragments in the track-side ditch is visible post-fall.**

The presence of multiple mobile fragments means that these events may be classified as fragmental rockfalls (Hungr and Evans, 1988; Hungr et al., 2014). Hungr and Evans (1988) originally proposed that for the case of fragmental falls, the movement of the most mobile fragments in the fall are independent of each other. This suggests that the modelling of these events as a volume of fractured material is unnecessary, and instead single design blocks with a specified average or maximum fragment volume could be used. This is in contrast with the idea that larger volume rock slope failures such as rock avalanches, should be modelled as granular flows rather than independent ballistic trajectories (Bourrier et al., 2013). The distinction between these two types of motion is often discussed in relation to the volume of material mobilized as part of the event, with larger volumes ($>10^3 - 10^4$ m³) suggested to show stronger interaction between individual fragments. A discussion of the





various volumes and classifications of rock slope failure relevant to the transition between these two styles of motion is presented by Corominas et al. (2017) and suggests that volume thresholds and terminology for these types of events is not yet consistent in the literature.

5      While rockfall events <1000 m$^3$, such as those considered in this study, may be described as having limited interaction between mobile fragments, the use of a single design block is not effective in cases where sufficiently large fragmental falls might overwhelm ditches at the base of a slope. A schematic overview of this process can be seen in Figure 9. Material which builds up in the ditch or other retaining structures may allow trailing rockfall debris to roll out over the newly formed surface. Additionally, a trailing fragment of rock may impact the accumulated pile of debris with enough force to push some fragments out onto the track. Simulation of the entire fragmental rockfall volume at once, as a moving mass of many fragments, allows

10    for important slope stopping features such as benches, gullies, and ditches to be filled, impacting the runout of trailing rockfall material. Snapshots from a video of a recorded fragmental rockfall which filled a ditch and impacted a section of railway in western Canada, can be seen in Figure 10. In this case, the leading portion of the rockfall event was pushed forward by trailing material before it fully came to rest in the ditch. Additionally, subsequent individual rock fragments were able to runout into the track region as a result of the catchment ditch being full.



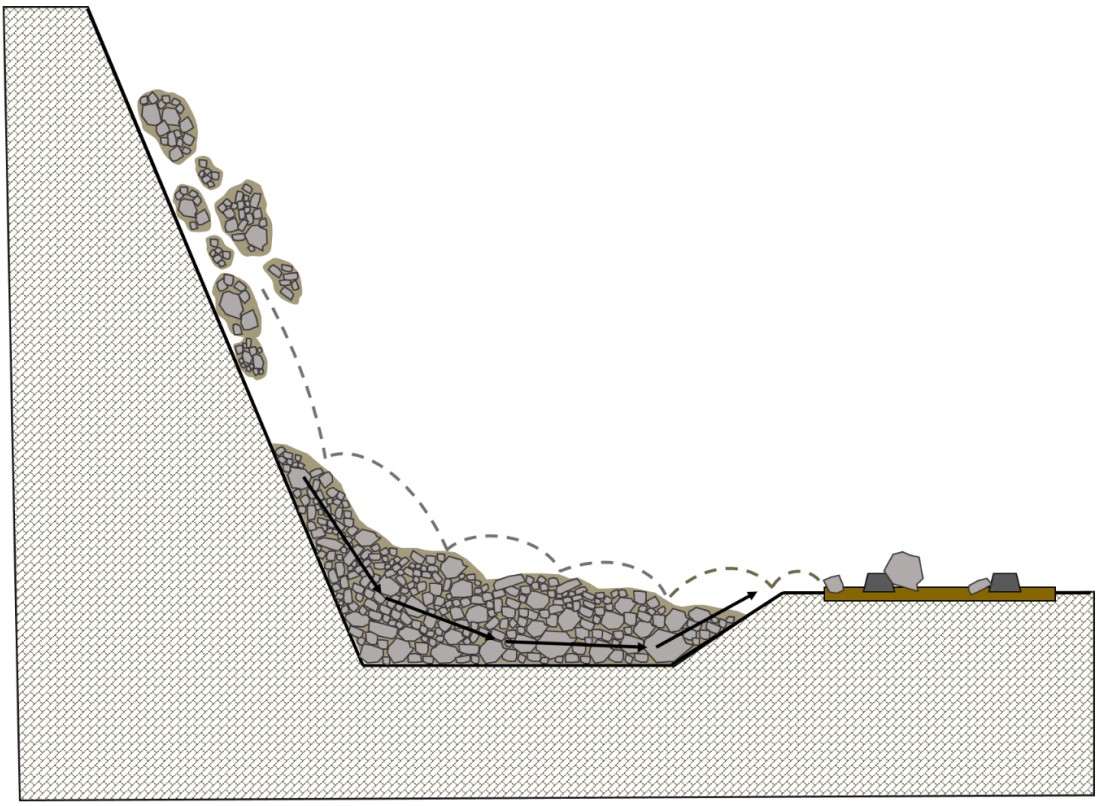

**Figure 9: Schematic of potential fragmental rockfall runout behaviour. Here leading material from the event has filled up a catchment ditch. Trailing rockfall material impacting the back of the deposit has the potential to push the leading material forward and out of the ditch (black arrows) reaching the track area. Additionally, the initial material has created a surface over which trailing rock fragments can move (grey trajectory), reducing the effectiveness of the ditch.**



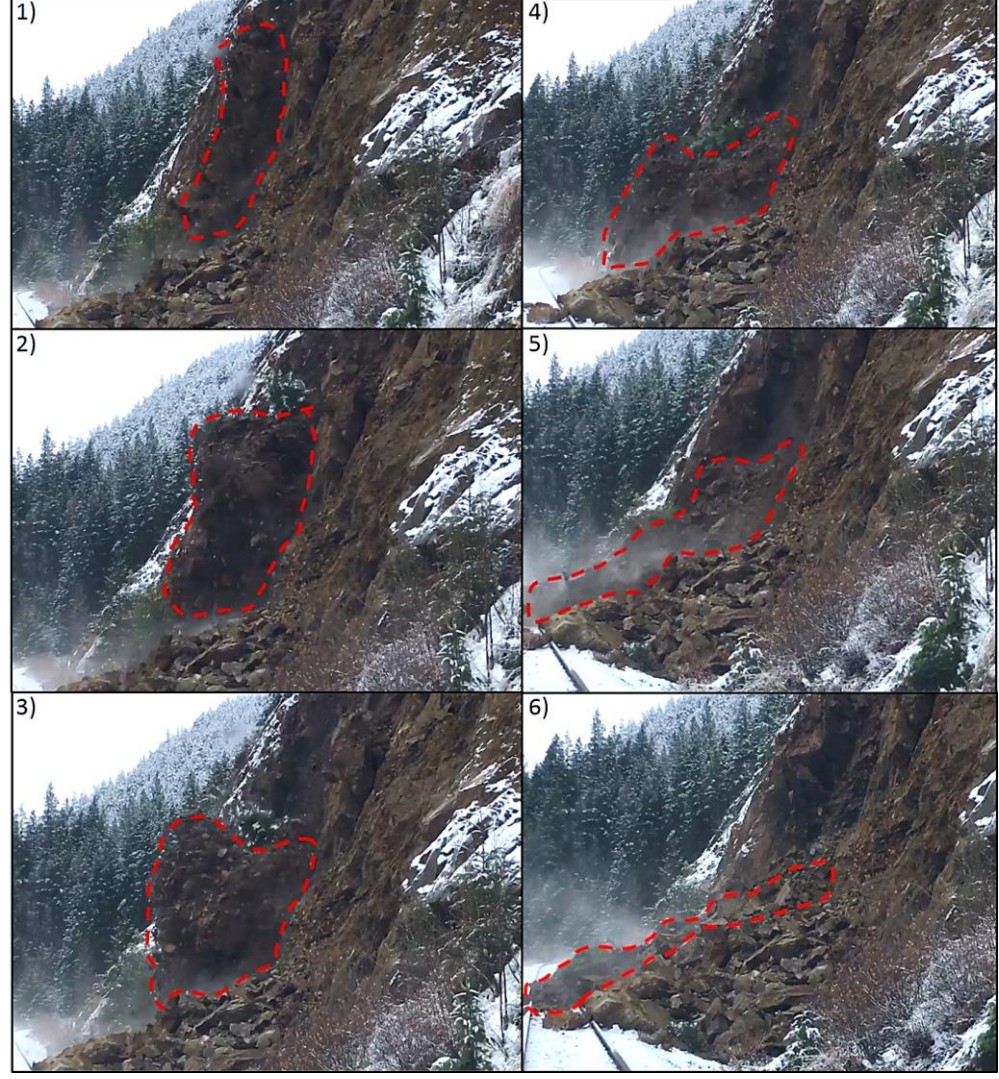

**Figure 10: Snapshots from a video of a fragmental rockfall event occurring above a section of railway in western Canada. Elapsed time between frame 1) and frame 6) is approximately 2 seconds.**

The simulation of these events as single block falls would produce unrealistic runout due to interaction with retaining structures like ditches. In addition, the mobility of larger falls is much different than smaller fragments, with significantly higher potential energies at release, as well as larger moments of inertia affecting rotation during runout. At track level the impact energy of a single coherent block would be larger, resulting in an overestimate of the potential for damage to the track.

5    Additionally, a single block simulation may underestimate the spatial extent of track interaction, as individual fragments from a multi-block fall are free to disperse from each other laterally, impacting multiple points on the track at once. From a design perspective, this would influence the type of mitigation selected for the site. The isolated high-energy impacts of a large single



block may require a fixed, high-strength system like a mechanically-stabilized earth wall or rock shed. In the case of the numerous, small impacts from a fragmented source volume, wire-mesh draping to divert the fragments into the ditch may suffice.

## 4 Runout Simulations

For each of the selected rockfall events from our field sites, we can set up a simulation in the Unity Game Engine. Each simulation is comprised of slope and rockfall geometry elements. The slope geometry is a 3D mesh produced from the pre-fall TLS scan of the slope, generated using Poisson Reconstruction in Cloud Compare. The rockfall geometry is a 3D mesh produced from the pre- and post-fall TLS scans of the source region, generated using Poisson Reconstruction, and connected into a single hull in Blender (Blender Foundation, 2017). This hull is then segmented into a collection of convex hull fragments,

using Voronoi fracturing in Blender. This method subdivides the source rockfall volume into a specified number of fragments selected by the user, with control over the size and shape of fragments produced. The end result is a simulated rockfall source volume which reflects the shape and fractured nature of the rockfall event detected from the field. A detailed explanation of the steps necessary to move from field remote sensing data to a functioning simulation in Unity, including the workflow from Cloud Compare to Blender to Unity, can be found in Sala (2018). One of the key strengths of this modelling technique is its

ability to simulate the collision between multiple moving rigid bodies at once. Support for multi-body collisions allows us to simulate the disaggregation of heavily jointed source rockfall volumes. This enables us to study the types of rockfall runout expected to occur at our field sites, taking into consideration the implications of fragmental runout discussed above. Snapshots of an example fragmental rockfall simulation for the 170 m$^3$ Goldpan event (Site A) can be seen in Figure 11.





**Figure 11: Screenshots of a fragmental rockfall simulation (1000 fragments) for the 170 m³ rockfall event at Goldpan.**

When developing or attempting to calibrate any new simulation technique, it is necessary to compare simulation outputs to real observations of the phenomena you are modelling in order to test how well the simulation performs. Where direct observation of rockfall events is possible, such as in rockfall drop tests, (e.g. Ushiro et al., 2006; Vick et al., 2015; Volkwein et al., 2018), this information could include translational and rotational velocity, mapped runout trajectories, or pass height above the slope. For the five rockfall cases selected from our field sites, this information is not available as the events were not observed directly. Instead the location and magnitude of change present in the change detection results for each event serve as the basis for simulation comparison.

Simulations of each of the five selected rockfall events were run and change maps of the simulation results were produced. In order to generate change detection maps for the simulations, the positions of the post-simulation fragments were merged





with the slope geometry to create a post-simulation 3D mesh. Similarly, the fragmented rockfall geometry, pre-simulation, was merged with the slope geometry to create a pre-simulation 3D mesh. Each mesh was then converted into a point cloud in Blender. The conversion of the mesh data to a point cloud allows us to use the M3C2 point-point distance calculation for both the actual and simulated rockfall events. The transition from simulation mesh to point cloud is done by selecting only visible

vertices in the mesh, using a similar vantage point to the direction of the original TLS scan. The selection of only visible vertices in the mesh is necessary as the 3D rockfall fragments, pre- and post-simulation, have vertices across their entire surfaces, both front and back facing. This results in multiple layers of points in the pre-fall source zone, and post-fall accumulation zone, adversely affecting point cloud normal calculations and producing errors in the distance measurements between what should be distinct pre- and post-fall surfaces. The extraction of visible vertices from the mesh geometry, using

a specified vantage point, ensures that only a single layer of points is present in the simulated point cloud.

The results of the change detection analysis for each simulated rockfall event are shown in Figures 12 – 16. In each case the game-engine based simulation prototype was able to produce rockfall runout which compared well in terms of the location and magnitude of the measured actual change. The percentages of source rockfall volume retained at different locations on the slope is shown. Volume measurements using the actual event point cloud data were completed using the 2.5D Volume tool in

CloudCompare. Simulated volume percentages were determined using the 3D mesh data for each of the convex hull fragments simulated in Unity3D. For each event, the simulated volume percentages for each of the slope locations was within 5% of the actual percentage, with the majority of the rockfall material ending up in the track-side ditches for every rockfall event except the Goldpan wedge slide. In the Goldpan case, Site A, we see two distinct areas of accumulation, on the mid-slope bench and on top of the rock shed. In this case the simulation results produced similar accumulations in both locations, with the majority

of the material leaving the slope over the edge of the rock shed.





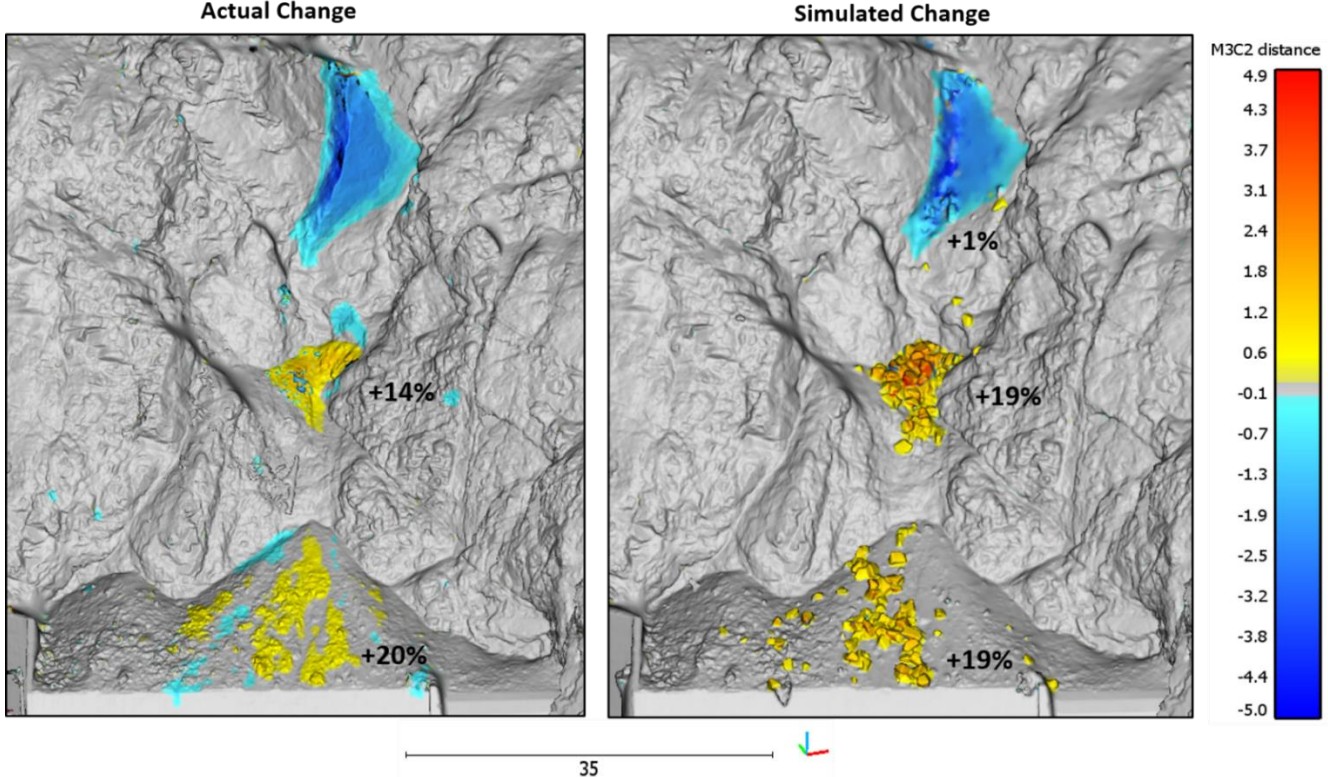

**Figure 12: A comparison of the actual and simulated change detection results from the 170 m³ wedge sliding event at Goldpan, Site A. The simulation produced a good comparison with the actual change results, yielding a similar spatial distribution, and magnitude of change. In both cases accumulation takes place on the mid-slope bench and on top of the rock shed, with the majority of material running out over the shed and off the slope. The percentage of initial source volume retained at various locations on the actual and simulated slope is displayed and shows a strong agreement. Areas of additional loss in the actual change detection, on top of the rock shed and at the mid-slope bench, are not captured in the simulation results. The simulation technique uses a fixed, rigid slope surface, and is not able to capture smaller slope failures which may have occurred as a result of talus impacts during the initial 170 m³ event.**




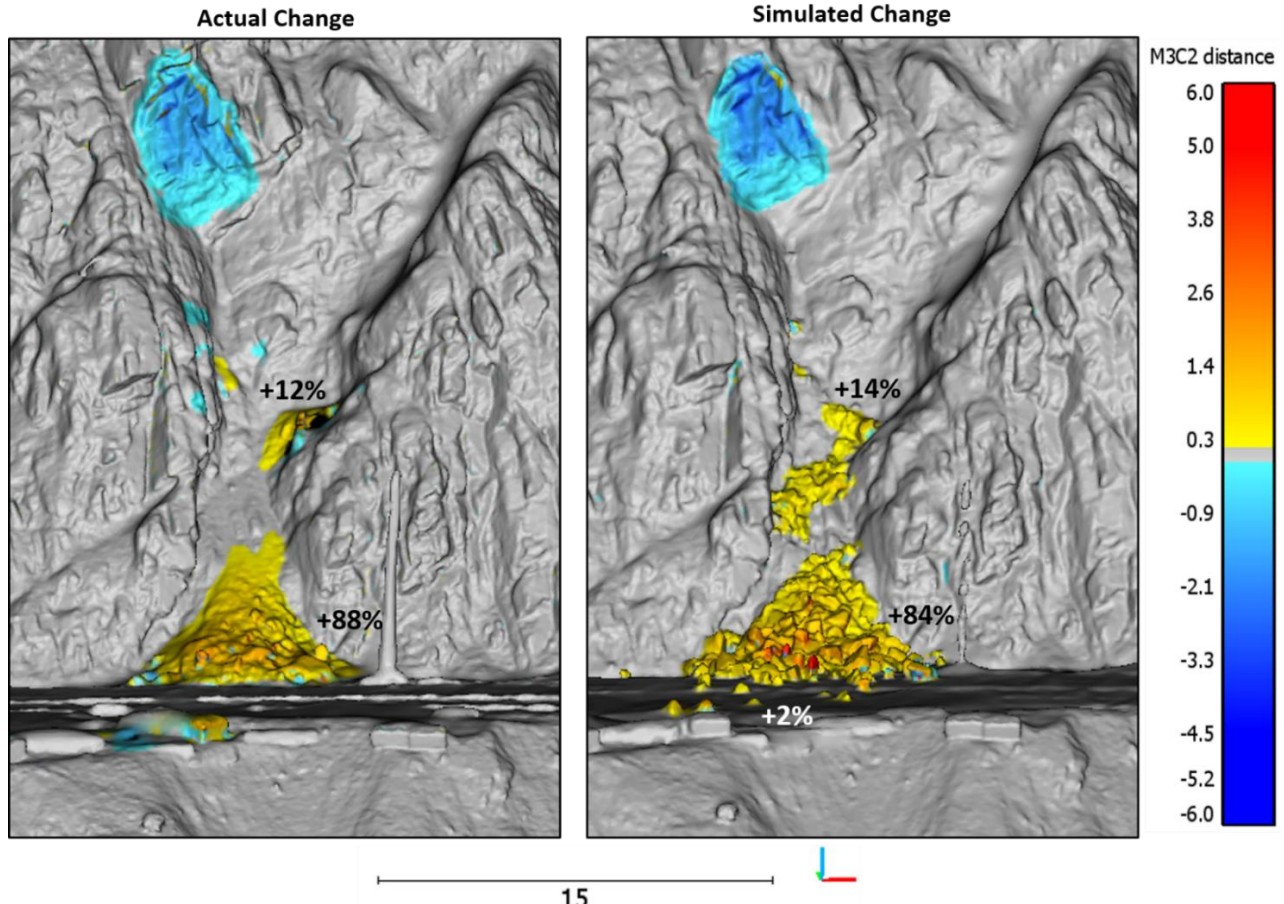

**Figure 13: A comparison of the actual and simulated change detection results from the 24 m³ wedge sliding event from White Canyon East, Site B. The simulation produced a good comparison with the actual change results, yielding a similar spatial distribution, and magnitude of change. The percentage of initial source volume retained at various locations on the actual and simulated slope is shown. In this case the majority of material in both the actual and simulated analysis ends up in the track-side ditch, with a smaller component of the volume stopped inside the gully. In the simulated change a small number of fragments do reach track level, representing approximately 2% of the simulated volume.**



**Figure 14: A comparison of the actual and simulated change detection results from the 32 m³ flexural toppling event from White Canyon East, Site C. The simulation produced a good comparison with the actual change detection results, yielding a similar spatial distribution, and magnitude of change. The percentage of initial source volume retained at various locations on the actual and simulated slope is shown. The majority of material in both cases is retained in the track-side ditch. In the simulated case a small amount of the simulated volume comes to rest on the slope (0.5%) and runs out into the track area (1.5%).**




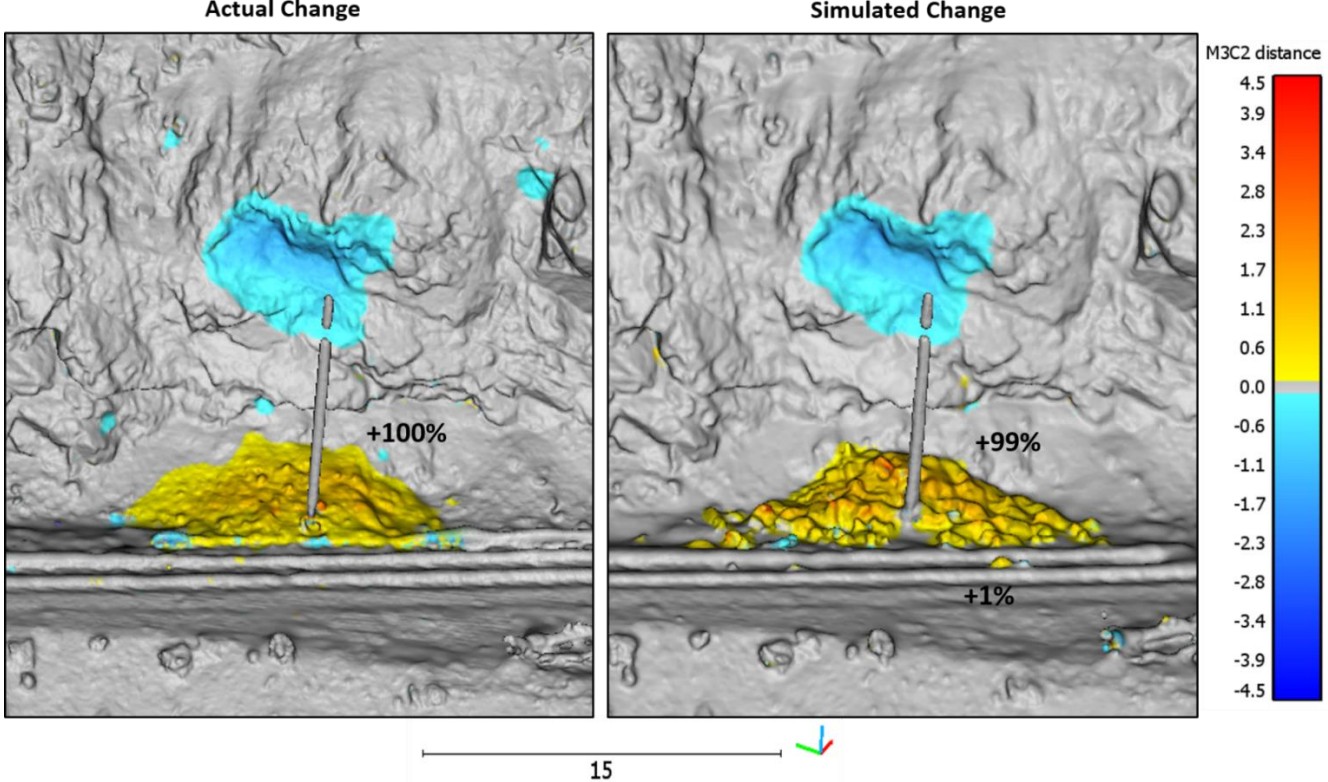

**Figure 15: A comparison of the actual and simulated change detection results from the 15 m³ planar sliding event from White Canyon West, Site D. The simulation produced a good comparison with the actual change detection results, yielding a similar spatial distribution, and magnitude of change. The percentage of initial source volume retained at various locations on the actual and simulated slope is shown. The majority of material in both cases is retained in the track-side ditch. In the simulated case, 1% of the volume reaches the track area.**



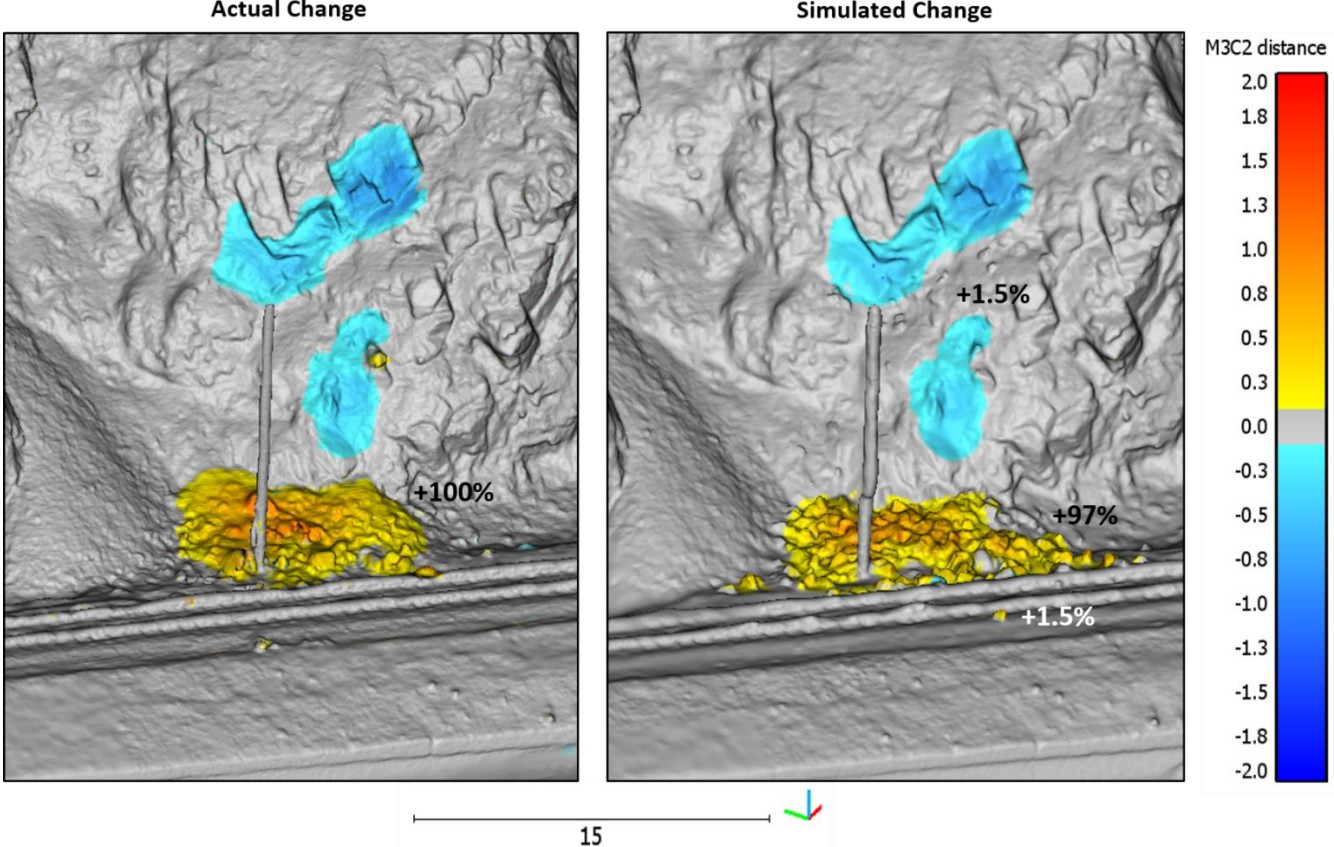

**Figure 16: A comparison of the actual and simulated change detection results from the 8 m³ planar sliding event from White Canyon West, Site E. The simulation produced a good comparison with the actual change detection results, yielding a similar spatial distribution and magnitude of change. The percentage of initial source volume retained at various locations on the actual and simulated slope is shown. The majority of material in both cases is retained in the track-side ditch. In the simulated case a small amount of the simulated volume comes to rest on the slope (1.5%) and runs out into the track area (1.5%).**

### 4.1 Simulation with Multiple Parameterizations

In each of the above change-based simulation comparisons, a single model iteration which produced a positive comparison to the observed change was shown. The input parameters used to produce each comparison are not identical, with variation between the five sites. For a single simulation, a parameter set consisting of coefficients of dynamic friction, restitution, and

5 viscoplastic ground drag, that best simulated the actual results was chosen. The parameter sets used for each of the five site comparisons are shown in Table 1.





**Table 1: The material coefficients used for each of the five rockfall event simulations. The fragmentation levels selected for the 15 simulation iterations run for the White Canyon East wedge sliding event are also included.**

|  | Coarse | Medium | Fine |
|---|---|---|---|
| Fragment # | 250 | 500 | 1000 |

| Site | Material | fric | rest | vpg |
|---|---|---|---|---|
| Goldpan Wedge Slide | rock | 0.5 | 0.4 | 0 |
|  | talus | 0.35 | 0.3 | 0.15 |
| White Canyon East Wedge Slide | rock | 0.45 | 0.25 | 0 |
|  | talus | 0.3 | 0.2 | 0.3 |
| White Canyon East Flexural Topple | rock | 0.55 | 0.25 | 0 |
|  | talus | 0.45 | 0.2 | 0.3 |
| White Canyon West Planar Slide 1 | rock | 0.5 | 0.3 | 0 |
|  | talus | 0.4 | 0.2 | 0.3 |
| White Canyon West Planar Slide 2 | rock | 0.45 | 0.3 | 0 |
|  | talus | 0.35 | 0.25 | 0.3 |

While a single parameter set can be selected producing results which align with the material accumulations seen in the change detection, we cannot be certain that this parameter set will best reflect conditions at any given site in the area. The friction, restitution, and damping coefficients used are a simplification of a suite of interconnected and complex processes which govern the loss of energy during collision between the rockfall and slope. For example, variations in moisture content

for a section of soil affects slope deformation and therefore the transfer of energy which takes place during collision (e.g. Vick, 2015). From a forward modelling perspective, while advances in rock-slope monitoring have shown that we are now able to detect pre-failure deformation for unstable sections of slopes (e.g. Royan et al., 2013; Kromer et al., 2015; Kromer et al., 2017), in some cases providing time-to-failure estimates, we are not able to predict the exact time at which a failure will occur. Therefore, in terms of material parameters, we cannot know the exact state of the slope when runout will take place. For this

reason, it makes sense that a range of potential material coefficients be used to capture the variability in these parameters.

Similarly, for forward modelling of events based on pre-failure deformation or pre-cursor rockfall, the exact volume of a failure would not be known prior to the event. While volume estimates based on structure in the source rock mass are possible (e.g. Lambert et al., 2012; e.g. Salvini et al., 2013), they require assumptions about joint persistence and shape. As a result, it would be advisable to estimate a range of potential volumes using the least and most conservative persistence estimates.

Additionally, while we may be able to identify potential failure events, and even estimate block volume from visible structure, we won't know the degree of fracturing expected as part of the disaggregation of the source volume. With this in mind, it is



important that we also use a range of fragmentation values as well, in this case termed coarse, medium and fine. The use of a suite of material coefficients, volumes, and levels of fragmentation, allows us to produce an envelope of potential rockfall runout rather than a single deterministic result.

An example suite of simulations has been produced for the 24 m$^3$ overhanging wedge failure at White Canyon East, Site B. Varying source volumes were not used in this case as the rockfall has already occurred, and the source volume is therefore known. The material coefficients used were based on the five parameter sets from Table 1, each of which produced a positive comparison at one of the five sites. These five material parameterizations were each run for three levels of fragmentation (250 fragments, 500 fragments, 1000 fragments) produced using Voronoi fracturing in Blender. The distribution of potential rockfall runout for the event was produced for each of these 15 simulations. Each point shown on the slope surface in Figure 17 is the end position of a fragment from one of the 15 simulations. The red points represent the runout of 95% of the simulated volume. This boundary illustrates that the majority of the material from the event, based on 15 different simulation parameterizations, is retained by the track-side ditch. This result compares well with the actual change detection from the event, in which all of the rockfall material appears to have come to rest in the gully above track or in the track-side ditch. The 99% (purple) and 100% (blue) runout boundaries are also shown, illustrating that a portion of the additional 5% of simulated material does come to rest in the track area.





**Figure 17: Results of the 15 simulations run for the 24 m³ wedge sliding event at White Canyon East, Site B. Each point represents the end point of a simulated fragment in one of the 15 simulations. The envelope of red points indicates the runout of 95% of the simulated volume, with the purple and blue portions representing envelopes containing the additional 4% and 1% of the material, respectively.**

## 4.2 Volume Comparisons and Ditch Design

From an engineering design perspective, one application of this simulation technique is the assessment of mitigation

performance. The simulation of a fragmental rockfall event as a single large block overestimates potential impact energies and



underestimates the spatial distribution of impacts possible when rockfall volumes runout as hundreds or thousands of individual, interacting fragments. The simulation of rockfall volumes as a collection of free-moving fragments allows us to produce more realistic material accumulations in retaining ditches adjacent to the track.

Using the 15 simulations run for the 24 m$^3$ White Canyon East wedge sliding event, Site B, we can analyze the distribution

5 of accumulated material along the slope surface. Figure 18 shows the same end point locations as in Figure 17 but in section view. Also included is a plot displaying the volume of material accumulated along the slope surface. From this distribution we observe a notable peak in volume occurring at the location of the track-side ditch. This style of volume accumulation curve allows us to visualize the effectiveness of countermeasures at retaining material. In preparation for an expected rockfall event, or during the construction of new ditches along a section of railway or highway, this analysis could be used to evaluate the

10 effectiveness of different ditch shapes, or the use of simple retaining structures like lock blocks. Additionally, the retention capacity and the effectiveness of countermeasures, as material progressively accumulates, could be tested.



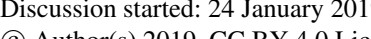

**Figure 18: A section view of the 15 simulation iterations run for the 24 m³ wedge sliding event at White Canyon East, Site B. The top plot shows the accumulation of simulated fragments across the section. The notable peak visible at 270 m is in alignment with the location of the track-side catchment ditch, as shown in the lower plot.**




In the initial 15 simulations run for the White Canyon East wedge sliding event, the pre-failure slope surface was used in order to examine the match between the simulated and actual conditions on the slope during runout. We can also re-run these 15 simulations using the post-failure slope surface. In this case, the ditch area is full of debris and large rockfall fragments from the 24 m$^3$ event, representing a ditch near full retention capacity. The simulation results using the post-failure surface

5    conditions can be seen in Figure 19, compared to the initial pre-failure analysis. From this we can see that the 95% volume runout boundary has moved closer to the track for the full ditch scenario, with some end points now touching the track boundary. A comparison of the two volume accumulation profiles for these different ditch scenarios can be seen in Figure 20. Two vertical lines indicate the location of the 95% volume-runout boundary for the two curves, with the red line indicating the start of the track area. The use of the post-failure, full ditch slope model, resulted in the 95% boundary moving forward 1.2 m,

10   illustrating a decrease in the effectiveness of the ditch at stopping material when full. From a back-analysis perspective this effect on runout illustrates the importance of using the true pre-failure slope conditions for our simulation comparisons. From a forward analysis perspective, it may not be possible to know the state of the ditch during the runout event, and therefore different ditch conditions should be simulated in order to assess the range of possible outcomes when evaluating countermeasure options.





Figure 19: Comparison of the 15 simulation iterations of the 24 m³ wedge sliding event at Site B using the pre-fall (empty) and post-fall (full) ditch geometries. In the full ditch scenario the 95% and 99% runout boundaries have shifted further forward into the track area, indicating a decreased effectiveness in the ditch at stopping material from reaching the track.

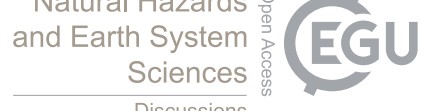

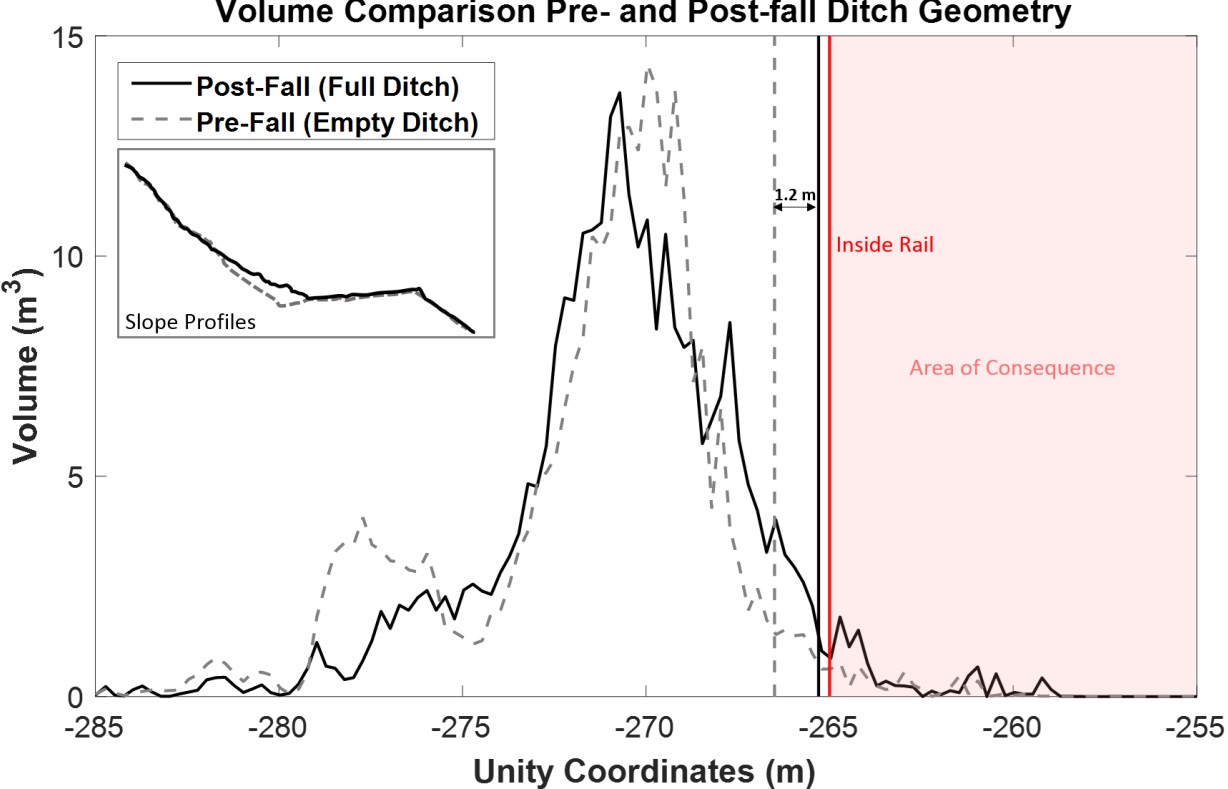

**Figure 20: The distribution of simulated volume for the pre-fall and post-fall ditch geometries is shown. The vertical grey and black lines indicate the 95% volume-runout boundaries for the two cases. Simulation using the post-fall (full) ditch geometry results in the 95% volume-runout boundary shifting 1.2 m closer to the inside rail.**

A comparison of the percentage of rockfall volume retained in different areas of the model for the empty and full ditch scenarios is shown in Figure 21. Simulation using the post-fall ditch geometry results in almost twice as much material reaching track level. Additionally, we can look at the size distribution of the fragments which end up in the track area, as shown in Figure 21 for the full ditch scenario. The use of volume-based runout envelopes takes into consideration both the size and number of fragments which reach the track. Different sized fragments have different implications from a hazard management perspective. For example, in the CN Rockfall Hazard Risk Assessment framework (Abbot et al., 1998a; 1998b), rockfall fragments with maximum dimensions of 0.3 – 1 m are noted as the sizes most likely to cause derailments due to their potential to get wedged underneath a train car. Of the 11% of simulated fragments which reach the track in the 15 simulations of the full ditch scenario, approximately 57% of them fall within the 0.3 – 1 m size range.




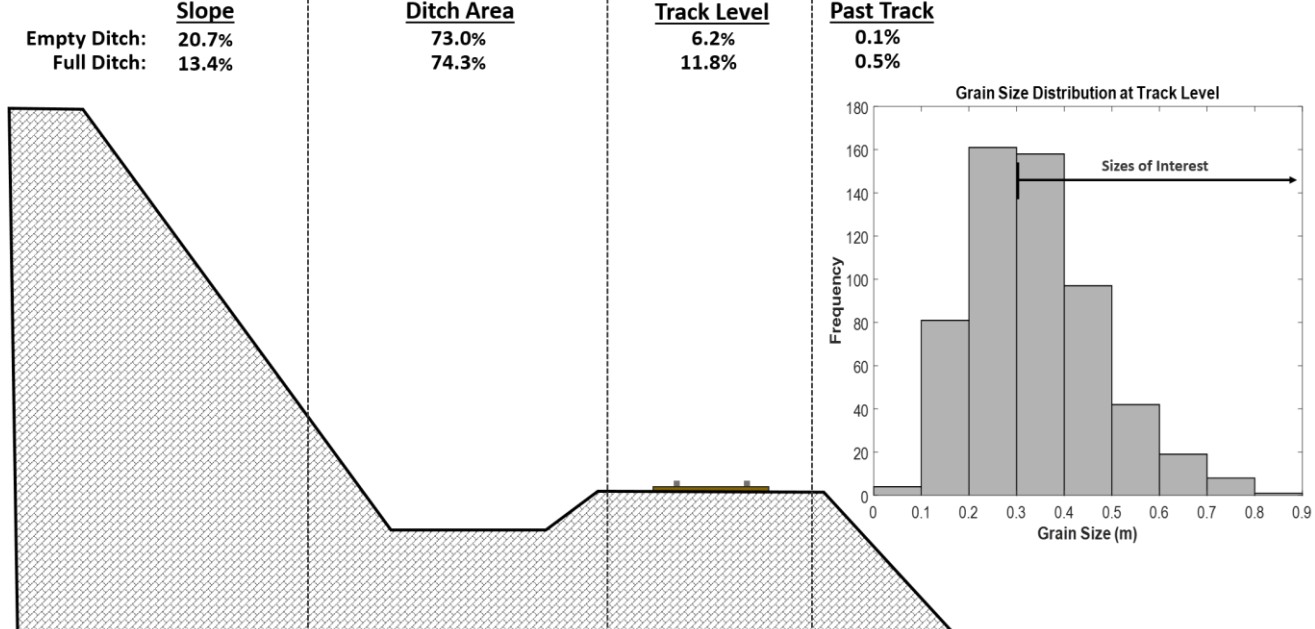

**Figure 21: A schematic overview of the volume percentages stopping at different locations for the 15 simulation iterations using the empty and full ditch geometries. Simulation using the full ditch geometry resulted in nearly twice as much material reaching track level. The distribution of the grain sizes which reached track level in the full ditch scenario is also shown.**

## 5 Limitations

The objective of this work was to demonstrate the ability of our rockfall simulation workflow to model complex rockfall source geometries comprised of many moving fragments. While this process is a step forward in the simulation of fragmental rockfall events, a discussion of current limitations is important. The two main limitations of this work are fragmentation timing and

5    the distribution and size of rockfall fragments used.

### 5.1 Fragmentation Timing

The events modelled in this paper were not observed directly and therefore it is not possible, using the available TLS data and site photographs, to understand at what point fragmentation took place during the event. While the post-fall accumulations visible in the photos and change detection indicate that the final state of the source volumes were deposits of coarse rock

10    fragments, the transition from coherent rockmass to thousands of fragments could have occurred immediately at the time of detachment, or as a result of impact during the first or subsequent collisions between the source volume and slope surfaces. The current methodology simulates the release of rock from the source zone as a gravity-induced, rapid unravelling of highly jointed material (Figure 22a). It is possible that this unravelling of jointed rockmass could have instead occurred slowly over



time as small isolated events (Figure 22b). Alternatively, the separation of the source mass into individual blocks could have occurred as a result of loading during impact (Figure 22c). At the same time, the decision to model the events as a rapid disaggregation of jointed rock was not arbitrary. This style of fragmentation is akin to the disaggregation-without-breakage rockfall mechanism described by Ruiz-Carulla et al. (2017), in which the rockfall body separates along discontinuities present

5   in the source rock mass, with no breakage of individual, discontinuity-bounded blocks. This appears to be the dominant behaviour in the railway rock slope video shown in Figure 10. This type of rockmass behaviour is also observed in low-stress underground excavations, in which structurally controlled failures of highly jointed or crushed rock will unravel from excavation ceilings under the influence of gravity (Palmstrom, 1995).

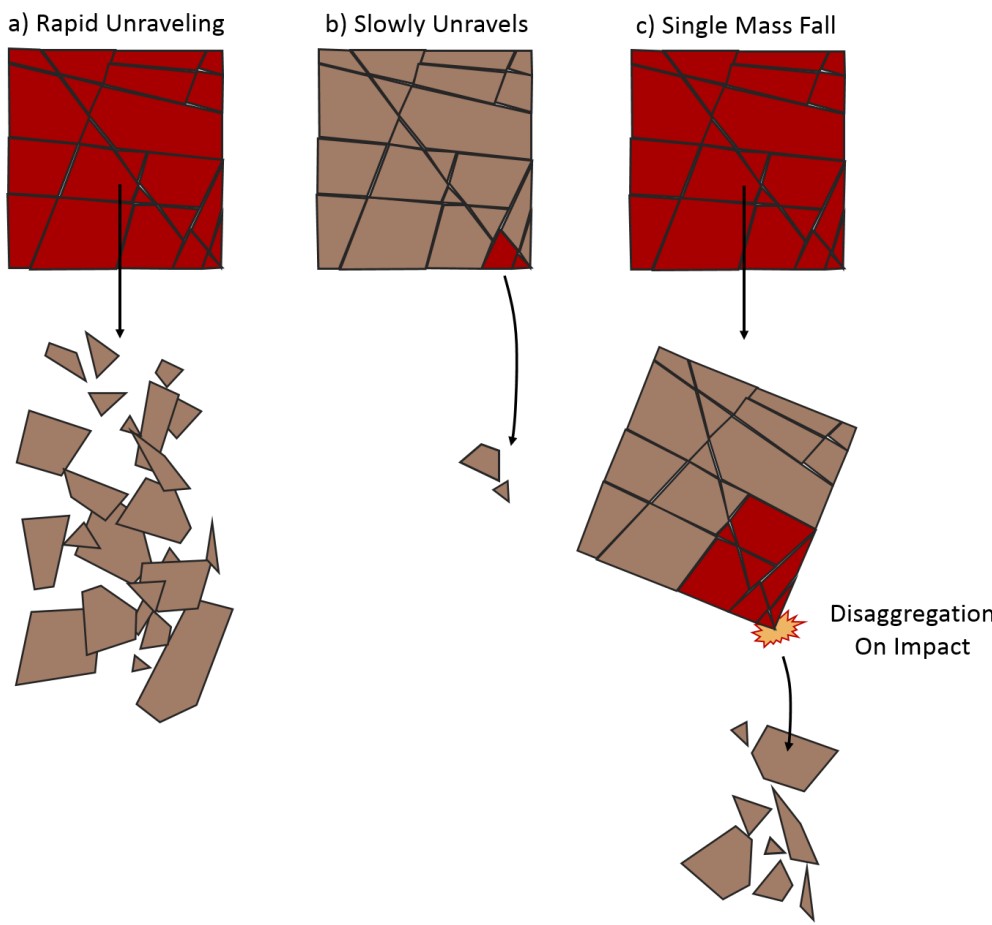

**Figure 22: Comparing potential timing of rockmass fragmentation, which could have taken place for the events discussed. Red block indicate material which breaks away from the initial source volume at a given time.**

10   The choice to model the events as a rapid unravelling of the entire source volume (Fig 22a) was also inherently influenced by the current capabilities of the modelling technique. In Sala (2018), the possibility to connect simulated fragments in the





source mass using strength-based object-to-object connections was briefly discussed. Each simulated connection between objects can be assigned a strength value, with the connection broken when this threshold is exceeded. The use of these connections could facilitate future simulations in which the source mass does not disaggregate immediately but instead as a result of force from impact. While this is one potential method for simulating break-up of the source volume during runout,

field data to support the calibration of these connections is not currently available.

## 5.2 Fragment Size Distribution

One of the goals of modelling rockfall source volumes as a collection of rock fragments was to better capture the size of mobile fragments involved in the rockfall events rather than using a single large block. While this effect was achieved using our modelling workflow, it should be noted that an evaluation of actual versus simulated grain size distribution has not been

completed.

At this time, Voronoi fracturing is used for source volume fragmentation. In 3D this method partitions the source mesh into disjointed convex polyhedra, based on seed points distributed in the mesh volume (Ledoux, 2007). Several parameters can be set in the fracturing algorithm including the recursive partitioning of larger fragments, and control over fragment shape. Detailed testing of these parameters has not yet been completed, but examples of recursive fracturing and shape effects can be

seen in Figure 23. While the use of this method is capable of rapidly creating fragmented 3D volumes, ready for use inside the Unity game engine, the fracture network produced is not based on actual discontinuities in the rock mass. For rock masses which are less jointed, discontinuity planes can be used in Blender to separate the source volume into blocks based on discrete joints mapped from available imagery or 3D models of the slope. In the case of the White Canyon and Goldpan events, where hundreds to thousands of fragments were visible in the post fall accumulations, the mapping of discrete joint planes was not

practical. For these cases the use of Voronoi fracture networks, instead of discrete joint planes, permitted us to model the sheer number of fragments present in the rockfall deposits.



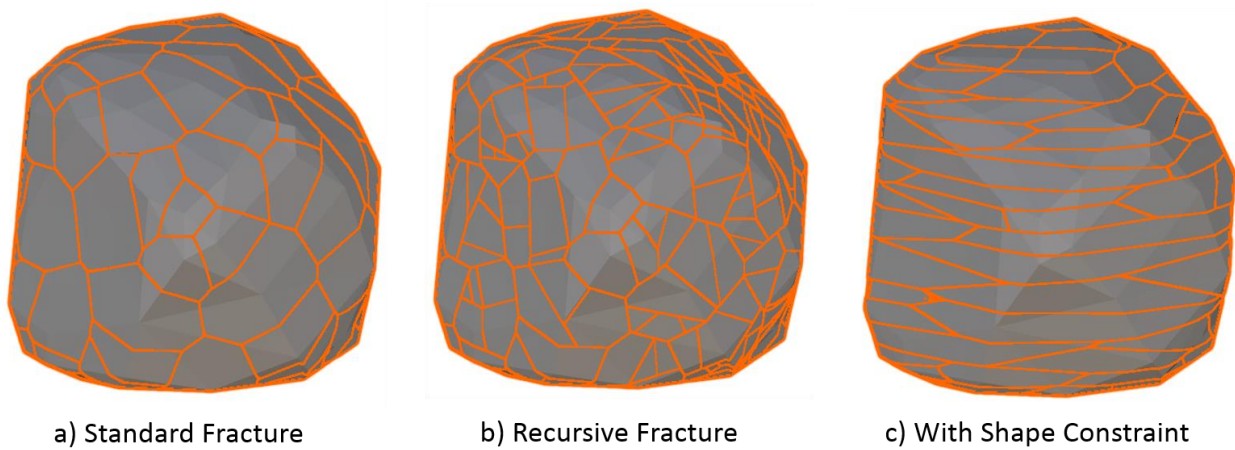

| a) Standard Fracture | b) Recursive Fracture | c) With Shape Constraint |

**Figure 23: Showing the results of different Voronoi fracturing methods. Recursive fracturing in b) results in the additional fracturing of the initial convex polyhedra in a). Shape constraints in c) result in the elongation of the convex polyhedra, creating a more sheet-like fracture network.**

The size of the fragments deposited during the five rockfall events are clearly not unimodal, with a variety of grain sizes present. Grain size distributions for these events were not created due to a lack of adequate data. Point spacing in the TLS data is too sparse to be able to isolate individual grains. Additionally, vantage points from our scan sites often result in notable occlusions at the track level, resulting in missing data in the post-fall debris piles. Photos of the events do capture the post-fall debris, but due to the oblique angle of the images, and homogenous colour of the material, accurate image-based grain segmentation is challenging.

A select few fragments measured for the White Canyon East 24 m$^3$ case, Site B, show approximate sizes ranging from 1.6 m to <0.05 m. Particles smaller than 5 cm are clearly present, but cannot be measured as the images are too noisy at this scale due to resolution limitations. Size ranges for the coarsest models (250 fragments) and finest models (1000 fragments) of the 24 m$^3$ event were 0.09 m – 0.8 m and 0.02 m – 0.6 m respectively. While we do not have a full grain size distribution for the actual event, we can see from these measurements that the grain sizes produced using the current Voronoi fracture parameters underestimate the largest block sizes and result in an overall smaller range of values. In terms of the smaller grains, the 2 cm minimum size of the 1000 fragment model is sufficient to capture the lower size limit of the 24 m$^3$ event. Ultimately more investigation needs to be done into fine-tuning the parameters of the Voronoi fracturing process in order to produce grain sizes with a broader range of values and that more accurately reflect the sizes produced by events in our study area. In order to do this, complete grain size distributions for events in in our rockfall database, and from other events in the literature, will need to be generated.



## 6 Conclusions

Five rockfall simulations were run based on TLS change detection inputs from events identified at our field sites in south-central British Columbia. The simulations utilized high-resolution 3D meshes generated from 6 – 10 cm TLS point clouds, and complex rockfall volumes, extracted from pre- and post-event scans. The fracturing of source rockfall volumes enabled

the modelling of fragmental rockfall runout, taking into consideration the interaction of fragments with each other as well as slope stopping features such as gullies and ditches. The results of this work demonstrate the ability of our rockfall modelling prototype to produce runout material accumulations which agree well with observed change from the actual events. With no direct observations of rockfall runout, or on-site measurement, the conversion of post-simulation mesh data in order to produce simulated change detection presents a novel way to perform rockfall runout comparisons in the absence of other data types.

The application of the technique to mitigation design was also discussed, demonstrating the potential of the model to be used for investigating the effectiveness of rockfall catchment ditches. This type of analysis, visualizing the accumulation of rockfall runout volume downslope, could also be applied to the assessment of other countermeasure options such as lock block retaining walls or rock sheds. The ability to perform these types of analyses is limited in other conventional rockfall modelling programs, as they are often able to simulate only a single moving block at a time. While these models are very useful for single

block falls, and regional runout assessments, they are unable to capture the build-up of rockfall material on the slope, which can affect the path of subsequent rockfall fragments and the effectiveness of retaining structures. The simulation of these falls as multiple moving bodies in our technique is able to capture this interaction between rockfall material and the finite capacity of the structures used for rockfall protection.

**Data Availability**

The underlying research data for this project cannot be made publicly available as they are commercially sensitive and permission must be granted by our industry partners to access the data. Sample code and descriptions of how to setup rockfall simulations using the Unity3D software can be found in Sala (2018).

**Author Contributions**

ZS was responsible for the development of the fragmental rockfall simulations and analysis of simulated and actual change detection results. DJH provided technical guidance on the framework of the study and the overall goals of the work in the context of railway hazard management. RH provided technical guidance on the development of the game-engine simulation and analysis techniques presented. ZS was responsible for manuscript preparation. DJH and RH reviewed and edited the

manuscript.

**Competing Interests**

The authors declare that they have no conflicts of interest.



**Acknowledgements**

This research work was supported by the Railway Ground Hazards Research Program, funded by CN Rail, Canadian Pacific and an NSERC CRD grant, and supported by Transport Canada and the Geological Survey of Canada. Special thanks to the

members of the Queen's RGHRP group who participated in the collection of the data used for this research; Trevor Evans of CN for field campaign support and expertise related to railway operations; and Dave Gauthier for regular discussion surrounding the technical and written components of this work.

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
