# Peer review of "Simulation of Fragmental Rockfalls Detected Using Terrestrial Laser Scans from Rock Slopes in South-Central British Columbia, Canada"

_Natural Hazards and Earth System Sciences, 2018_

## Referee Comment (RC1) · Anonymous Referee #1 · 20 Feb 2019

This paper is well written and describes a well executed study of rockfall hazards near railroad tracks. It uses a combination of airborne and terrestrial lidar data and benefits from a high number of repeat data collections. Overall, the intent of the study, data collection and methods are all well described. Further information on a few points, listed below in question form, would improve the paper.

1. Page 4 Indicates that monitoring at two sites has been continuous since 2012, with scans taking place approximately every three months. How many TLS scans were used in this study? (ie. Data was collected from 2012 to what year for this study?)

2. Were the TLS and ALS scans taken concurrently (or nearly so?) such that they could

be compared for bias? (ie. Subtracting the TLS terrain, at a decimated resolution, from the ALS terrain should produce a near-zero result.)

---

## Short Comment (SC1) · 27 Feb 2019

The manuscript is a very interesting contribution. It presents the simulation of the trajectories of fragmental rockfalls. The authors use a Unity3D game engine that works with polyhedra and takes into account the shape of the fragments and the collision between them. The procedure is promising and may have immediate practical application in rockfall risk management. The manuscript is well organized and presented. I have, however, a few comments:

1) From a formal point of view, the references must be checked (e.g. page 2: Jaboyed-off et al is 2010 or 2012?; page 2: Sala et al. 2018; page 5: Struik 1987; Monger and

Nokelberg, 1996; among others, are not found in the reference list; Kramer et al. 2018 is not cited in the paper).

2) The discussion on the existing conventional rockfall models (page 2, lines 30-33) suggests that there are no codes able to model fragmental rockfalls, which is not correct. Nowadays some 3D rockfall runout models are able to simulate the disaggregation of the falling rock masses defined by pre-existing discontinuities (Cuervo et al. 2015) or by the breakage of the initial blocks upon impact on the ground surface (Wang and Tonon, 2011; Matas et al. 2017). This point should be clarified here and in the conclusions (page 37 lines 13-14).

Cuervo S, Daudon D, Richefeu V, Villard P, Lorentz J (2015) Discrete Element Modeling of a Rockfall in the South of the "Massif Central", France. In: Lollino G. et al. (eds) XII IAEG Congress. Engineering Geology for Society and Territory - Volume 2: 1657-1661. Springer

Matas G, Lantada N, Corominas J, Gili JA, Ruiz-Carulla R, Prades A (2017) RockGIS: a GISbased model for the analysis of fragmentation in rockfalls. Landslides 14: 1565–1578

Wang Y, Tonon F (2011) Discrete element modeling of rock fragmentation upon impact in rock fall analysis. Rock Mech Rock Eng 44:23–35

3) Section 4. Runout simulation (page 17, lines 12-14). Despite the authors mention that the steps of the simulation are described in Sala 2018, a minimal description is required here. In particular: the parameters used to run the model; whether it works with a single block or with an In-Situ Block Size Distribution; and the criteria for generating the fragments and their size-distribution.

4) The performance of the model is tested by fitting the simulation to the studied events (section 4.1). As acknowledged by the authors the set of parameters used to fit the model may not be the one that reflects the conditions at any given site in the area. In

10.5194/nhess-2018-321

order to perform forward modeling and highlight the value of the work, a guidance on how to determine the initial fracture pattern of the rock mass and the material coefficients should be included in the paper.

5) The model simulates the disaggregation of the falling rock mass but (apparently) not the breakage of blocks. Please, discuss how it may affect the results.

6) Finally, the simulated runout does not consider the size of the fragments. However, the intensity of the impact and the efficacy of the rockfall mitigation measures is affected by the size the falling blocks. Please discuss this point and how it should be taken into account in risk management.

––––––––––––––––––––––––––––––––

---

## Referee Comment (RC2) · Anonymous Referee #2 · 11 Mar 2019

Dear editor, Here is the review of the paper:

Simulation of Fragmental Rockfalls Detected Using Terrestrial Laser Scans from Rock Slopes in South-Central British Columbia, Canada

By Zac Sala, D. Jean Hutchinson, Rob Harrap

This paper is related to rockfall simulations based on game engine, in order to simulate blocks interactions and fragmentations for small rockfalls. Using 5 cases studies the paper demonstrates that after calibration of the model, it reaches results very close to the observation by LiDAR.

[Figure]

This paper is a novel and valuable contribution to the rockfall hazard modelling and representation. I am confident it will be a nice contribution for NHESS. Before making my comments, I read the comments of Corominas and I fully agree with him, I will not repeat some of his comments that are relevant. I support that the authors follow his advices.

I will be interesting to have cross-sections of the 5 sites it is easier to follow rockfall paths. Please explain on each figure what is the small arrow near the scale, or remove it (I guess it is the orientation?)

- References are missing such as Sala (2018), Ledoux etc..., but Sala is it accessible.

- At the end of the introduction about fragmentation please cite: Matas et al. and Ruiz-Carulla et al.

- Pages 4 line 3: I do not understand why 0.3 m, this a very poor resolution, contrast with line 11, see also page 37 line 2.

- P 17 line 14: it I really important that at least the model is described in a few lines and how plays the different parameters (half page is enough).

- Table 1: the parameters have to be included in the description requested above.

- P 26 lines7-8: no structure is used?

Figure 23: I think that this must be located in a method section with the related text..

---

## Author Comment (AC1) · 4 May 2019

Thank you for taking the time to review our paper. We appreciate the feedback, and kind words regarding our manuscript and applicability of our study.

1) Monitoring started at the site as early as 2012. Regular scans have been collected since 2014 and typically occur seasonally (approx. every 3 months). In addition to this seasonal scanning, some auxiliary data collection campaigns have collected scans at higher frequencies (daily – weekly) to support correlation with precipitation events, as well as look at smaller scale debris movements in debris channels on slope. The imagery and TLS data used for the rockfall events in this paper were from 2015 and

2016.

2) While there was overlap in the TLS and ALS data acquisitions, only two ALS acquisitions have taken place. They have not been compared for bias in this case, instead the ALS has been used mainly as a regional dataset for sections of railway corridor where we don't regularly collect TLS data. Additionally, the ALS has been used in certain studies to fill in holes in the data where incidence angle and vegetation results in occlusions in the TLS data.

---

## Author Comment (AC2) · 4 May 2019

Thank you very much for taking the time to read our paper Dr. Corominas. I found your work on rockfall fragmentation to be very informative when carrying out my Master's research on the subject. Your feedback here will be very helpful in revising the content of the manuscript.

1)&2) The reference list will need to be adjusted to correct the discrepancies you highlighted. Thank you for the suggested additional references on the topic of existing models incorporating fragmentation. We will add these references and additional text in the introduction (page 2) and (page 37) for clarification.

[Figure]

Suggested change – Update reference list and in-text citations so that they are consistent. Include additional reference material and text for clarification on existing rockfall models which include fragmentation on pages 2 and 37.

3) A description of the inner workings of the model was left out for concision due to the current length of the paper, and its existing description in the complete master's thesis document (Sala 2018). A brief description could be added for more clarity. The collision parameters used in the model currently are a friction coefficient, restitution coefficient, and viscoplastic dampening coefficient. The fracturing of the source rock uses a voronoi fracturing algorithm in the open source 3D modelling software Blender. There are many parameters which you can modify in this algorithm to achieve different results. The simplest method is specifying the number of fragments you would like, which generates a random fracture network in the 3D volume with that number of fragments of varying sizes and shapes. Additional parameters such as the aspect ratio of the fragments can be specified to simulate preferential orientation of fractures in the network. Without detailed information on the pre- and post-fall block size distribution, and this being a first trial using the fracturing algorithm, trial and error was used during simulation, running a suite of simulations with varying input parameters, namely the initial number of fragments. As suggested in the limitations section of the manuscript on pages 35-36, we would like to do a more detailed study of the block size distributions for these rockfall in order to calibrate our use of the fracturing algorithm in Blender.

4) Given that this modelling technique is still in its infancy, guidelines for picking initial input parameters (friction, restitution, viscoplastic dampening, source fragmentation) are still being studied. The technique should be tested on more case studies, with a variety of potential material types prior to a formal set of input guidelines being published. Testing at two additional study slopes, one in New Zealand, and one in Japan, is presented in Sala 2018, but is outside the scope of this paper. Given a particular set of inputs may not reflect the conditions at any given site when running forward models, we would advise to not treat the model as a discrete approach and instead run a

suite of simulations with a range of initial inputs. This allows you to look at the range of potential impact points, and the areas with the highest concentration of deposited material in the modelling results.

5) Additional discussion of the disaggregation vs. breakage concept can be added to the paper. As discussed in the paper, the nature of scanning the slopes infrequently means that we do not observe the rockfall event directly. While modelling the blocks as a mass of individual mobile fragments of different sizes and shapes, aligns with the material visible in the rockfall deposits, we can't be sure which scenario from Figure 22 actually takes place. If breakage of individual fragments was taking place, with new fractures being generated in each block, we would expect this to affect the amount of energy present in the rockfall system, as energy is consumed and released in the fracturing and breakup of the fragments. As is discussed in Sala 2018, there is potential with this game engine platform to specify connections between individual fragments in the source volume. These simulated connections have break forces, torques, etc. While the initial fracture pattern, and thus smallest fragments would still be pre-defined, this would allow for the incorporation of some strength-based "breakage" of the initial simulated volume. This is just proof-of-concept at the moment, as it was outside of the scope of the study, and in general requires calibration data that we do not have for these rockfall events. Nevertheless it is something we are interested in pursuing.

Suggested change – additional discussion of the disaggregation versus the breaking of intact rock fragments and how that impacts rockfall runout. Also include a comment on how our model has the potential to incorporate this in future development

6) I'm not entirely sure what you mean by the runout does not consider the size of the fragments. Each fragment that is moving, moves as a result of collisions with the slope and other fragments. This collision takes into consideration the 3D hull of each fragment, and therefore it's runout is affected by the fragments size. Similarly, the change maps produced post simulation highlight changes between the initial and post-simulation surfaces, which is affected by the size/volume of the individual simulated

fragments. Perhaps you are referring to the deposition points on Figure 17/19. These points do not reflect the individual size of the thousands of fragments that were simulated. While this could be changed, the decision to not do this was due to the fact there are so many deposition points over the range of simulations that meaningfully distinguishing size on this figure would be lost on the viewer.

---

## Author Comment (AC3) · 4 May 2019

Thank you for taking the time to review our paper. We appreciate the feedback, and appreciate the kind words about our manuscript. We agree that Dr. Corominas provided excellent feedback, and intend to follow his advice on the manuscript.

-The small arrow at the bottom of the figures showing the 3D point cloud and/or mesh data is the orientation of the 3D object in space. It is not critical to the figure, but is a default feature in Cloud Compare, the point cloud manipulation software being used.

-Thank you for pointing out the missing reference, this will be fixed. Similarly we can

add the suggested references Matas et al. and Ruiz-Carulla et al. at the end of the introductory discussion of fragmentation.

Suggested Change – Include additional references from Reviewer 2 and fix inconsistencies between reference list and in-text references

-The 0.3 m (30cm) point resolution listed is for the Airborne Laser Scans (ALS). This scanning technique typically yields lower resolution datasets when compared to the terrestrial methods used to collect the 6-10 cm point clouds for the case studies of the five individual rockfall events. This difference in resolution between the techniques is due to a number of factors including the lidar unit used to collect the data, stationary vs. mobile data collection, and the distance between the scanning platform and the slope. -A detailed description of the model was left out for concision as it exists in the full master's thesis document (Sala, 2018). A brief description of the input parameters in table 1 (friction coefficient, restitution coefficient, viscoplastic ground drag coefficient) and high level overview of the model could be included here for clarity, similar to the recommendation by Dr. Corominas.

Suggested change – Provide a brief overview of the model physics including a description of each of the model parameters

-I agree that Figure 23 could be included earlier in the manuscript when initially discussing the Voronoi fragmentation technique used for running these simulations. I think this would help the reader visualize what is happening, and provide more context for the discussion to come.

Suggested change – Move Figure 23 up to Page 17, likely between lines 10 – 15 for additional clarity on the Voronoi process earlier in the paper

---

## Author Response (AR2)

**nhess-2018-321 Author Replies to Reviewer Comments**

**Reply to Reviewer 1**

The below comments were left in the public discussion replying to Reviewer 1. Based on the focus of the manuscript, existing explanation of the field data in the manuscript, and Reviewer 1's report specifying no edits required, no changes were made in the manuscript for these comments.

**Comment:** Page 4 Indicates that monitoring at two sites has been continuous since 2012, with scans taking place approximately every three months. How many TLS scans were used in this study? (ie. Data was collected from 2012 to what year for this study?)

**Reply:** Monitoring started at the site as early as 2012. Regular scans have been collected since 2014 and typically occur seasonally (approx. every 3 months). In addition to this seasonal scanning, some auxiliary data collection campaigns have collected scans at higher frequencies (daily – weekly) to support correlation with precipitation events, as well as look at smaller scale debris movements in debris channels on slope. The imagery and TLS data used for the rockfall events in this paper were from 2015 and 2016.

**Comment:** Were the TLS and ALS scans taken concurrently (or nearly so?) such that they could C1 NHESSD Interactive comment Printer-friendly version Discussion paper be compared for bias? (ie. Subtracting the TLS terrain, at a decimated resolution, from the ALS terrain should produce a near-zero result.)

**Reply:** While there was overlap in the TLS and ALS data acquisitions, only two ALS acquisitions have taken place. They have not been compared for bias in this case, instead the ALS has been used mainly as a regional dataset for sections of railway corridor where we don't regularly collect TLS data. Additionally, the ALS has been used in certain studies to fill in holes in the data where incidence angle and vegetation results in occlusions in the TLS data.

**Reply to Reviewer 2**

The below comments were left in the public discussion replying to Reviewer 2. Changes made based on a reviewer comment are indicated below the comment.

**Comment:** Please explain on each figure what is the small arrow near the scale, or remove it (I guess it is the orientation?)

**Reply:** The small arrow at the bottom of the figures showing the 3D point cloud and/or mesh data is the orientation of the 3D object in space. It is not critical to the figure, but is a default feature in Cloud Compare, the point cloud manipulation software being used.

**Comment:** - References are missing such as Sala (2018), Ledoux etc. . ., but Sala is it accessible. At the end of the introduction about fragmentation please cite: Matas et al. and Ruiz-Carulla et al.

**Reply:** Thank you for pointing out the missing reference, this will be fixed. Similarly we can add the suggested references Matas et al. and Ruiz-Carulla et al. at the end of the introductory discussion of fragmentation.

**Change Made:** Included additional references and discussion of previous fragmentation work on pages 2-3. Also fixed inconsistencies between reference list and in-text references.

**Comment:** Pages 4 line 3: I do not understand why 0.3 m, this a very poor resolution, contrast with line 11, see also page 37 line 2.

**Reply:** The 0.3 m (30cm) point resolution listed is for the Airborne Laser Scans (ALS). This scanning technique typically yields lower resolution datasets when compared to the terrestrial methods used to collect the 6-10 cm point clouds for the case studies of the five individual rockfall events. This difference in resolution between the techniques is due to a number of factors including the lidar unit used to collect the data, stationary vs. mobile data collection, and the distance between the scanning platform and the slope.

**Comment:** P 17 line 14: it I really important that at least the model is described in a few lines and how plays the different parameters (half page is enough). Table 1: the parameters have to be included in the description requested above.

**Reply:** A detailed description of the model was left out for concision as it exists in the full master's thesis document (Sala, 2018). A brief description of the input parameters in table 1 (friction coefficient, restitution coefficient, viscoplastic ground drag coefficient) and high level overview of the model could be included here for clarity, similar to the recommendation by Dr. Corominas.

**Change Made**: Provided an overview of the model physics including a description of the impact coefficients on pages 17-18.

**Comment**: Figure 23: I think that this must be located in a method section with the related text..

**Reply:** I agree that Figure 23 could be included earlier in the manuscript when initially discussing the Voronoi fragmentation technique used for running these simulations. I think this would help the reader visualize what is happening, and provide more context for the discussion to come.

**Change Made**: Moved Figure 23 to page 18 for additional clarity on the Voronoi process earlier in the paper. The figure is now Figure 11.

**Reply to Jordi Corominas**

The below comments were left in the public discussion replying to Jordi Corominas. Changes made based on a reviewer comment are indicated below the comment.

**Comment:** From a formal point of view, the references must be checked (e.g. page 2: Jaboyedoff et al is 2010 or 2012?; page 2: Sala et al. 2018; page 5: Struik 1987; Monger and C1 NHESSD Interactive comment Printer-friendly version Discussion paper Nokelberg, 1996; among others, are not found in the reference list; Kramer et al. 2018 is not cited in the paper).

The discussion on the existing conventional rockfall models (page 2, lines 30-33) suggests that there are no codes able to model fragmental rockfalls, which is not correct. Nowadays some 3D rockfall runout models are able to simulate the disaggregation of the falling rock masses defined by pre-existing discontinuities (Cuervo et al. 2015) or by the breakage of the initial blocks upon impact on the ground

surface (Wang and Tonon, 2011; Matas et al. 2017). This point should be clarified here andin the conclusions (page 37 lines 13-14).

**Reply:** The reference list will need to be adjusted to correct the discrepancies you highlighted. Thank you for the suggested additional references on the topic of existing models incorporating fragmentation. We will add these references and additional text in the introduction for clarification.

> **Change Made:** Updated reference list and in-text citations so that they are consistent. Included additional reference material and text for clarification on existing rockfall models which include fragmentation on pages 2-3.

**Comment:** Section 4. Runout simulation (page 17, lines 12-14). Despite the authors mention that the steps of the simulation are described in Sala 2018, a minimal description is required here. In particular: the parameters used to run the model; whether it works with a single block or with an In-Situ Block Size Distribution; and the criteria for generating the fragments and their size-distribution.

**Reply:** A description of the inner workings of the model was left out for concision due to the current length of the paper, and its existing description in the complete master's thesis document (Sala 2018). A brief description could be added for more clarity. The collision parameters used in the model currently are a friction coefficient, restitution coefficient, and viscoplastic dampening coefficient. The fracturing of the source rock uses a voronoi fracturing algorithm in the open source 3D modelling software Blender. There are many parameters which you can modify in this algorithm to achieve different results. The simplest method is specifying the number of fragments you would like, which generates a random fracture network in the 3D volume with that number of fragments of varying sizes and shapes. Additional parameters such as the aspect ratio of the fragments can be specified to simulate preferential orientation of fractures in the network. Without detailed information on the pre- and post-fall block size distribution, and this being a first trial using the fracturing algorithm, trial and error was used during simulation, running a suite of simulations with varying input parameters, namely the initial number of fragments. As suggested in the limitations section of the manuscript on pages 36-37, we would like to do a more detailed study of the block size distributions for these rockfall in order to calibrate our use of the fracturing algorithm in Blender.

> **Change Made:** Provided an overview of the model physics including a description of the impact coefficients on pages 17-18.

**Comment:** The performance of the model is tested by fitting the simulation to the studied events (section 4.1). As acknowledged by the authors the set of parameters used to fit the model may not be the one that reflects the conditions at any given site in the area. In C2 NHESSD Interactive comment Printer-friendly version Discussion paper order to perform forward modeling and highlight the value of the work, a guidance on how to determine the initial fracture pattern of the rock mass and the material coefficients should be included in the paper.

**Reply:** A particular set of inputs may not reflect the conditions at any given site when running forward models. As a result we would advise to not treat the model as a discrete approach and instead run a suite of simulations with a range of initial inputs. This allows you to look at the range of potential impact points, and the areas with the highest concentration of deposited material in the modelling results,

based on a suite of input parameters. An example of this is shown in Figure 18, with discussion on page 27, lines 4 to 15.

Given that this modelling technique is still in its infancy, guidelines for picking initial input parameters (friction, restitution, viscoplastic dampening, source fragmentation) are still being studied. The technique should be tested on more case studies, with a variety of potential material types prior to a formal set of input guidelines being published. Testing at two additional study slopes, one in New Zealand, and one in Japan, is presented in Sala 2018. Meaningful discussion of these results is outside the scope of this paper, as these case studies could be an entirely separate paper themselves.

**Comment:** The model simulates the disaggregation of the falling rock mass but (apparently) not the breakage of blocks. Please, discuss how it may affect the results.

**Reply:** As discussed in section 5.1, fragmentation timing, the rockfall events in this paper were not observed directly. The nature of scanning the slopes infrequently means that we do not observe the rockfall event when it occurs. While modelling the blocks as a mass of individual mobile fragments of different sizes and shapes, aligns with the material visible in the rockfall deposits, we can't be sure which scenario from Figure 23 actually takes place. If breakage of individual fragments was taking place, with new fractures being generated in each block, we would expect this to affect the amount of energy present in the rockfall system, as energy is consumed and released in the fracturing and breakup of the fragments. As is discussed in lines 1 to 5 on page 36, there is potential with this game engine platform to specify connections between individual fragments in the source volume. These simulated connections have break forces, torques, etc. While the initial fracture pattern, and thus smallest fragments would still be pre-defined, this would allow for the incorporation of some strength-based "breakage" of the initial simulated volume. This is just proof-of-concept at the moment, as it was outside of the scope of the study, and in general requires calibration data that we do not have for these rockfall events. Nevertheless it is something we are interested in pursuing.

**Comment:** Finally, the simulated runout does not consider the size of the fragments. However, the intensity of the impact and the efficacy of the rockfall mitigation measures is affected by the size the falling blocks. Please discuss this point and how it should be taken into account in risk management.

**Reply:** I'm not entirely sure what is meant by the runout does not consider the size of the fragments. Each fragment that is moving, moves as a result of collisions with the slope and other fragments. This collision takes into consideration the 3D hull of each fragment, and therefore it's runout is affected by the fragments size. Similarly, the change maps produced post simulation highlight changes between the initial and post-simulation surfaces. This is inherently affected by the size/volume of the individual simulated fragments, as the simulated block shape is part of the mesh generated post-simulation. Perhaps you are referring to the deposition points on Figure 18-20. These points do not reflect the individual size of the thousands of fragments that were simulated. While this could be changed, the decision to not do this was due to the fact there are so many deposition points over the range of simulations that meaningfully distinguishing size on this figure would be lost on the viewer. The grain size distribution reaching the track region in Figure 22, is an example of how the size of each simulated fragment can be used, and would not be possible if we did not consider the size of the fragments during runout.

[revised manuscript text omitted]

**Commented [ZS11]:** Added per reviewer comments